

# Monopole breaking of Chern-Weil symmetries

Eduardo García-Valdecasas[1,2], Matthew Reece[3] and Motoo Suzuki[3,4]

**1** SISSA, Via Bonomea 265, Trieste 34136, Italy
**2** INFN, Sezione di Trieste, Via Valerio 2, 34127, Italy
**3** Jefferson Physical Laboratory, Harvard University, Cambridge, MA 02138, USA
**4** KEK Theory Center, Tsukuba 305-0801, Japan

## Abstract

Gauge theories in $d$ dimensions with a nontrivial fundamental group admit a $(d-3)$-form magnetic symmetry and a $(d-5)$-form instantonic symmetry. These are examples of Chern-Weil symmetries, with conserved currents built out of the gauge field strength, which can only be explicitly broken through violations of the Bianchi identity. For U(1) gauge theory, it is clear that magnetic monopoles violate not only the $(d-3)$-form magnetic symmetry but also lower-form symmetries like the instantonic symmetry. It is also known that an improved instanton number symmetry current, which is conserved, can be constructed in the case that the magnetic monopole admits a dyonic excitation. We study the generalization to other gauge groups, showing that magnetic monopoles also violate instantonic symmetries for nonabelian groups like PSU($n$), and that dyon modes can restore such symmetries. Furthermore, we show that in many (but not all) examples where a gauge group $G$ is Higgsed to a gauge group $H$, the structure of monopoles and dyons emerging from the Higgsing process explicitly breaks the instantonic symmetries of $H$ to those of $G$. The meaning of explicit breaking of a $(d-5)$-form symmetry is clearest for $d > 4$, but these results also extend to $d = 4$, where the breaking is interpreted as an obstruction to coupling the theory to a background axion field.

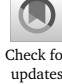

# 1 Introduction

## 1.1 Setting

Quantum field theories can admit a wide variety of symmetries, which provide a nonperturbative tool for understanding dynamics. Recent years have seen great progress in understanding new kinds of generalized symmetries in higher-dimensional QFT, building on the viewpoint of [1, 2] (for reviews, see, e.g., [3–8]). In this paper, our focus will be on a particular class of symmetries, arising in gauge theory, for which the symmetry charge is a topological invariant of the gauge bundle. These invariants are known as *characteristic classes* in mathematics, and we will follow [9] in referring to the corresponding symmetries as *Chern-Weil symmetries*.

Examples include symmetries that measure magnetic flux or instanton number. Some related work includes [10–20].

The conservation law for Chern-Weil symmetries follows from the Bianchi identity in gauge theory, which makes such symmetries difficult to break explicitly. As emphasized in [9], this poses a puzzle for quantum gravity, which is not believed to admit global symmetries [21–26]. This principle implies that, when a quantum gravity theory admits a low-energy description in terms of gauge theory, all of the Chern-Weil symmetries of the gauge theory must be either gauged or explicitly broken. Chern-Weil symmetries can be gauged in a straightforward manner by coupling their symmetry currents to dynamical gauge fields via Chern-Simons terms. This may help to explain why Chern-Simons terms are ubiquitous in known theories of quantum gravity (see also [27]). Explicit breaking is more difficult to understand: because the symmetry follows from the topology of the gauge group, we cannot simply add explicit symmetry breaking terms to a Lagrangian. In this paper, we aim to provide a clearer understanding of how magnetic monopoles can explicitly break Chern-Weil symmetries in a variety of gauge theory examples.

We consider a gauge theory with gauge group $G$ in $d$ spacetime dimensions. We will focus on the scenario where $G$ is compact and connected. We will be primarily interested in two symmetries associated with the topology of $G$. The first symmetry of interest, the *magnetic $(d-3)$-form symmetry*, measures the magnetic flux of the gauge theory and acts on 't Hooft operators [2]. The $(d-3)$-form symmetry group is given by $\pi_1(G)^\vee$, the Pontryagin dual of the fundamental group of $G$. For example, the group U(1) has fundamental group $\mathbb{Z}$, so U(1) gauge theory has a $\mathbb{Z}^\vee \cong$ U(1) $(d-3)$-form symmetry, with symmetry charge the magnetic flux $\frac{1}{2\pi} \int_\Sigma F \in \mathbb{Z}$ measured on any closed 2-surface $\Sigma$.

The second symmetry of interest, the $(d-5)$-*form instanton number symmetry*, has an integer charge which is proportional to the integral $\int_X \text{tr}(F \wedge F)$ over a 4-manifold $X$. In $d > 4$ spacetime dimensions, this is a generalized global symmetry. In the special case $d = 4$, this is not a symmetry in the usual sense, but is what we might refer to as a "$(-1)$-form global symmetry." This can be alternatively understood as the possibility of coupling the theory to a background axion field $\theta \cong \theta + 2\pi$ [28–30] (see also [31–34]). We will be concerned with how the instanton number symmetry can be explicitly broken in the case $d > 4$, and in how a coupling to a background axion field might be obstructed in the case $d = 4$. We will refer to the latter case as explicit breaking of the $(-1)$-form global symmetry.

## 1.2 Magnetic symmetry gauging or breaking

Gauging and breaking of the magnetic $(d-3)$ form symmetry is well understood. Consider first the case of U(1) gauge theory, where the symmetry current is $\frac{1}{2\pi}F$. We can gauge the symmetry by coupling the current to a dynamical $(d-2)$-form gauge field $B$. This $\frac{1}{2\pi}B \wedge F$ term gives the U(1) gauge field a mass, removing it from the low energy spectrum. On the other hand, the $(d-3)$-form symmetry can be explicitly broken by the introduction of dynamical magnetic monopoles, which modify the Bianchi identity to take the form $dF = 2\pi J_{\text{mag}}$ where $J_{\text{mag}}$ is the magnetic monopole current (which can also be thought of as a three-dimensional delta function localized to the monopole's worldvolume).

In the more general case, when the fundamental group of $G$ contains a discrete factor, gauging the discrete $(d-3)$-form magnetic symmetry corresponds to changing the global structure of the gauge group. For example, PSU$(n) \cong$ SU$(n)/\mathbb{Z}_n$ gauge theory has a $(d-3)$-form $\mathbb{Z}_n$ symmetry. Gauging this symmetry produces an SU$(n)$ gauge theory. While this operation does not have as dramatic an impact on low-energy physics as in the U(1) case, it does change the spectrum of Wilson and 't Hooft lines in the theory [35]. SU$(n)$ gauge theory admits more electrically charged matter representations than PSU$(n)$ gauge theory, and in quantum gravity, we expect that corresponding charged particles exist.

Explicit breaking of the discrete $(d-3)$-form symmetry is always accomplished by introducing magnetic monopoles.[1] In the PSU($n$) example, these monopoles carry a $\mathbb{Z}_n$ magnetic charge.

## 1.3 Instantonic symmetry gauging or breaking

The instanton number symmetry can be gauged by coupling the theory to a dynamical $(d-4)$-form gauge field $C$ through a Chern-Simons term of the form $C \wedge \mathrm{tr}(F \wedge F)$. In some cases, the symmetry can also be broken by the introduction of appropriate dynamical objects. In U(1) gauge theory, the instanton symmetry current $J_{\mathrm{inst}} = \frac{1}{8\pi^2} F \wedge F$ is simply the wedge product of two magnetic symmetry currents; as a result, as soon as $\mathrm{d}F \neq 0$, we have $\mathrm{d}J_{\mathrm{inst}} \neq 0$ as well (for $d > 4$).[2]

Much of this paper is concerned with generalizing this result to other symmetry groups $G$. At first glance, one may think that a magnetic monopole charge valued in $\mathbb{Z}_n$ cannot contribute to the breaking of an instanton number charge valued in $\mathbb{Z}$ (such a statement was made in [9]). However, this is not the case. Again, the gauge group PSU($n$) provides a prototype. Relative to SU($n$), the instanton number can take fractional values proportional to $1/n$, and the fractional part of the PSU($n$) instanton number is determined by the square of the $\mathbb{Z}_n$-valued class $w_2$ that determines the magnetic monopole number [36]. (A detailed table of such relationships for different groups can be found in [29].) In this way, magnetic monopole breaking of the $(d-3)$-form symmetry with current $w_2$ can induce breaking of the $(d-5)$-form instanton number symmetry, as we will discuss in detail in Sec. 4.

Another well-known element of the story for the U(1) case is that, when the magnetic monopole admits an appropriate dyonic collective coordinate and can acquire electric charge [37,38], an improved instanton number symmetry is preserved by the monopole [9,39]. Whether such a dyonic mode is present depends on details of the UV completion; for example, it is present for the 't Hooft-Polyakov monopole. We will see that a generalization of this result applies to a much wider class of examples. When a gauge group $G$ is higgsed to a subgroup $H \subset G$, the IR theory in general has more instantonic symmetries than the UV theory. For example, breaking SU(5) to the Standard Model gauge group $G_{\mathrm{SM}} \cong [\mathrm{SU}(3) \times \mathrm{SU}(2) \times \mathrm{U}(1)]/\mathbb{Z}_6$ leads to three distinct instanton number symmetries in the IR, one for SU(3), one for SU(2), and one for U(1). It was explained in [9] that the two independent linear combinations of these symmetries that are not inherited from the SU(5) instanton number symmetry are explicitly broken by a process that shrinks the instanton below the higgsing scale, rotates it within SU(5), and blows it back up. In this paper we will see that there is a different description of this breaking in terms of magnetic monopoles. The minimal $G_{\mathrm{SM}}$ monopole carries flux under all three gauge group factors, and so at first glance would appear to fully break all of the instanton number symmetries. We will see that properly accounting for its dyon mode allows us to construct an improved current for precisely one linear combination of these symmetries, which is the one inherited from SU(5). We show that this generalizes to a wide variety of examples: magnetic monopoles, together with their dyonic zero modes, provide a defect in the low-energy effective field theory that remembers the Chern-Weil symmetries of the ultraviolet theory. However, we will also see examples that do not fit this pattern, where the infrared gauge group does not admit magnetic monopoles.

---

[1]A canonical example is the SU(2) Georgi-Glashow model where 't Hooft-Polyakov monopoles break the magnetic symmetry of the IR U(1) gauge theory.

[2]This result clearly carries over to lower-form symmetries with current $F \wedge F \wedge F \wedge \cdots$, though we will not discuss such cases in detail.

## 1.4 Instantonic symmetry in $d = 4$

There is a longstanding claim that string theory has no free parameters.[3] The string coupling constant, for example, arises as the vacuum expectation value of a dynamical dilaton field. This is the prototype for many examples in which parameters are determined by the values of moduli fields.

The claim that quantum gravity has no free parameters suggests that, for a 4d gauge theory appearing in the low-energy EFT of a quantum gravity theory, the $\theta$ angle should not be a continuously tunable parameter. Instead, either it is the value of a dynamical axion field $\theta(x)$, or some principle will restrict it to take on only a discrete set of possible values. Both of these cases are realized in the context of known string theory vacua. Axion fields are ubiquitous in constructions of particle physics models within string theory (see, e.g., [41–43]). On the other hand, in some cases there is no light modulus field controlling the $\theta$ angle. For example, in a Type IIB compactification on a rigid Calabi-Yau, the $\theta$ angle for the 4d gauge field obtained by reducing $C_4$ on the holomorphic 3-cycle can only take the CP-conserving values $\theta = 0$ or $\theta = \pi$ [44]. Similar results have been found in flux compactifications [45]. A field-theoretic analogue is the result that in many 3d QFTs, the fractional part of Chern-Simons contact terms in the two-point function of conserved currents is fixed by general principles [46]. These contact terms are holographically dual to $\theta$ angles in 4d AdS quantum gravity. A similar result holds for the would-be $\theta$ angle of the $\mathbb{CP}^1$ sigma model in $(2+1)$d [47].

Viewing a $(-1)$-form U(1) global symmetry of a theory as a means of coupling the theory to a background $\theta$ angle, we can interpret these two possibilities in the language of symmetry. When there is a dynamical axion $\theta(x)$ coupled to gauge fields, we say that the would-be $(-1)$-form global instanton number symmetry of the theory has been gauged.[4] On the other hand, when some physics obstructs the coupling to a background $\theta(x)$, we say that the $(-1)$-form instanton number symmetry is explicitly broken. We expect that in this case, only discrete values of the $\theta$ parameter are attainable.

One reason that we find the language of $(-1)$-form symmetries to be useful in this context, rather than sticking to the more familiar language regarding absence of free parameters, is that the analogy to higher $p$-form symmetries and their explicit breaking proves to be a useful guide to possible obstructions to coupling to an axion. We have seen that magnetic monopoles break instanton number symmetry in $d > 4$. In $d = 4$, magnetic monopoles obstruct the coupling to a background axion field, due to the Witten effect [48]. Furthermore, just as in $d > 4$, the existence of a dyon mode on the monopole allows for the construction of an improved symmetry current to which the background axion can be consistently coupled [9, 39, 49]. Our results regarding the generalization of monopole breaking of instanton number symmetry, and dyon restoration of the UV instanton number symmetry in examples of higgsing $G \to H$, all carry over in this way to the case $d = 4$.

## 1.5 Phenomenological motivation

The absence of global symmetries in quantum gravity provides a potentially powerful tool for making predictions about particle physics. For example, QED has a magnetic 1-form global symmetry. Of the two options for eliminating this symmetry described in §1.2, the first (gauging) is not viable in the world around us, because it would decouple the photon from the low-energy spectrum. The remaining option is that the symmetry is explicitly broken, and magnetic monopoles exist. This is a special case of the completeness hypothesis for quantum

---

[3]This is mentioned as "string lore" in [40], for example. We are unsure what the earliest published version of this claim is. A recent discussion using the language of $(-1)$-form symmetries appears in [32].

[4]The instanton number symmetry can also be gauged by coupling to massless chiral charged fermions [9, 30]. In both cases, this renders the current $J_{\text{inst}}$ exact.

gravity, which holds that all allowed gauge charges must be carried by dynamical objects in the theory [25, 26, 50–54]. The existence of magnetic monopoles is an interesting prediction of quantum gravity for the real world, but a difficult one to test experimentally, because they are likely to be very heavy.

The lack of a free $\theta$ parameter in quantum gravity, or equivalently the absence of a $(-1)$-form instanton number symmetry, may connect more directly to tractable experiments. One possibility is that the $\theta$ parameter is promoted to a dynamical axion field that gauges the $(-1)$-form symmetry. Such fields are the target of many ongoing experiments. The alternative is that some underlying principle fixes the $\theta$ parameter to a discrete set of possible values. This approach could also play a role in solving the Strong CP problem, as in Nelson-Barr models [55, 56], where CP symmetry requires that the fundamental $\theta$ parameter is zero (and spontaneous breaking of CP generates a small correction). It is interesting that, in the simplest examples of quantum gravity theories in which an instanton number symmetry is explicitly broken, there are no chiral fermions charged under the gauge field (various examples illustrating this point have been collected in [57, 58]).

To assess whether quantum gravity predicts that a dynamical axion field exists in our universe, coupled to the Standard Model gauge fields, or whether it allows for alternative means of removing the free parameter $\theta$ from the theory, we should classify the possible physical principles that can fix the value of $\theta$ (or, equivalently, explicitly break the $(-1)$-form global instanton number symmetry). This work constitutes a step toward this goal, by correcting and extending the incomplete discussion of magnetic monopole breaking of instanton number symmetry in earlier work [9]. However, as we will comment in the concluding section, more work remains to fully classify the ways in which instanton number symmetry can be explicitly broken.

## 1.6 Outline

The outline of this paper is as follows. In Sec. 2, we offer some general remarks about the explicit breaking of topological symmetries in the presence of dynamical topological defects. In Sec. 3, we review how dynamical magnetic monopoles explicitly break the Chern-Weil symmetries of U(1) gauge theory, and how a localized dyon mode on the monopole can be used to construct an improvement of the Chern-Weil symmetry current that is still conserved. Whether or not such a dyon mode exists depends on the UV completion. In Sec. 4, we show that this story extends to the case of $\mathrm{PSU}(n) \cong \mathrm{SU}(n)/\mathbb{Z}_n$. There is a $\mathbb{Z}_n$ magnetic $(d-3)$-form symmetry which is broken by the existence of dynamical monopoles. We argue that these monopoles also break the U(1) instantonic $(d-5)$-form symmetry, and that there is again a prospect of restoring the symmetry with dyons. In Sec. 5, we consider a range of examples where a simple and simply connected gauge group $G$ is higgsed to a smaller group $H$. In general, this leads to magnetic monopoles associated with $\pi_2(G/H)$ that break the magnetic $(d-3)$-form symmetries of $H$. These monopoles also break instanton number $(d-5)$-form symmetries associated with $H$. In certain cases, we show that the dyon modes on the monopoles allow the construction of improved instantonic symmetry currents that restore precisely the UV instantonic symmetry of $G$. However, in some cases we find that the IR theory has additional emergent instantonic symmetries that are not broken by magnetic monopoles. In Sec. 6, we offer some concluding remarks. Additional technical details are provided in two appendices. In App. A, we review the construction of the dyon effective action for the 't Hooft Polyakov monopole in the case of SU(2) → U(1), and give a further example of the dyon effective action for the case SU(3) → SO(3). In App. B, we discuss magnetic and instantonic symmetries in some additional examples of $G \to H$ higgsing patterns for which $G$ is not simple.

## 2 Explicit breaking of topological symmetries

Topological symmetries have charges that are conserved off-shell; discussing their explicit breaking in an EFT setting can only be done semiclasscaly. As an example consider a low energy theory in a 4d spacetime $M_4$ featuring a U(1) gauge field with a magnetic $U(1)_m^{(1)}$ symmetry. This symmetry follows from the Bianchi identity of the gauge field and is, therefore, conserved off-shell. As a result it can't be broken by a mild modification of the Lagrangian. It can only be broken by modifying the UV. A simple way is to consider this EFT as arising in the IR of an SU(2) gauge theory that undergoes adjoint Higgsing. This theory contains finite energy configurations carrying magnetic charge, the 't Hooft Polyakov monopoles. An 't Hooft line can end on the core of such a monopole and the magnetic charge is screened, thus breaking the magnetic $U(1)_m^{(1)}$ symmetry. From the deep IR these configurations look singular, but they have a finite energy set by the Higgsing scale. In the following we wish to be agnostic about the UV and assume that finite energy configurations carrying magnetic charge exist. We can define an 't Hooft line operator $H(\gamma)$ as an operator that excises an $S^2$ around a curve $\gamma$ and prescribes boundary conditions around it such that a magnetic charge is measured. The effect of an insertion of such operator in a correlation function is to restrict the field configurations to be included in the path integral to those satisfying,

$$\frac{1}{2\pi}dF = \delta^{(3)}(\gamma),\tag{1}$$

where $\delta^{(3)}(\gamma)$ is a 3-form which is Poincaré dual to $\gamma$. It is apparent from Equation (1) that summing over $H(\gamma)$ insertions (i.e., treating the magnetic monopoles as dynamical objects in the theory) explicitly breaks the $U(1)_m^{(1)}$ symmetry. If a probe 't Hooft line is introduced, its magnetic charge will be screened by the $H(\gamma)$ insertions. More generally, operators that can't be written in the electric variables of the theory are called magnetic or defect operators. In the following we introduce toy models with compact scalar fields where one can easily visualize the explicit breaking of the symmetry.

### 2.1 A compact scalar in 2d

Consider a theory of a compact scalar with period $2\pi$ in a Euclidean 2d space $M_2$,

$$S = \int \frac{1}{2}|d\phi|^2\,.\tag{2}$$

Here and elsewhere, $|d\phi|^2$ denotes $d\phi \wedge \star d\phi$. There is an electric $U(1)_e^{(0)}$ shift symmetry $\phi \to \phi + \alpha$, $\alpha \in [0, 2\pi)$. This symmetry implements a U(1) rotation on the gauge invariant[5] vertex operators $V(x) = e^{i\phi(x)}$. There is also a magnetic $U(1)_m^{(0)}$ symmetry under which monodromy defect operators are charged.[6] A monodromy defect operator $M(x)$ is defined by excising a point $x$ and prescribing a boundary condition $\frac{1}{2\pi}\int d\phi = 1$ around it.[7] We can explicitly break the $U(1)_m^{(0)}$ symmetry by summing over insertions of $M(x)$. Each combination of such insertions will correspond to including field configurations on a spacetime with a given number of points $x_i$ removed and prescribed monodromy charges emanating from them. For a single insertion $M(x)$, the allowed field configurations will satisfy

$$\frac{1}{2\pi}d(d\phi) = \delta^{(2)}(x),\tag{3}$$

---

[5]Note that $\phi \to \phi + 2\pi$ is a gauge redundancy.

[6]Equivalently, the magnetic symmetry acts by constant shifts on the dual boson $\tilde{\phi}$.

[7]A monodromy defect operator can also be inserted by prescribing a branch-cut singularity on the gauge variant field $\phi$.

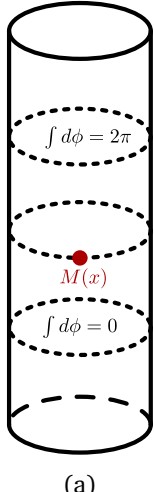

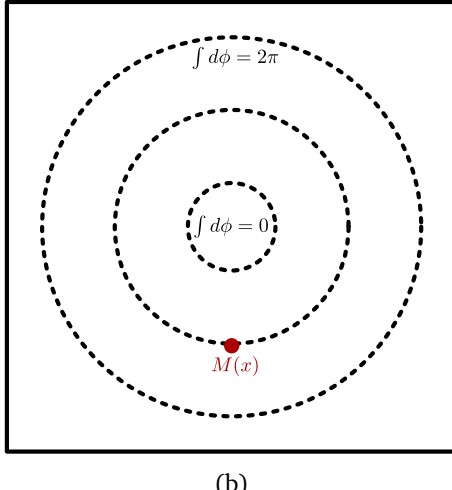

(a)             (b)

Figure 1: Compact scalar cylinder quantization with a monodromy operator. a) represents the theory on the cylinder. The quantization is along the circle and, as $x$ is crossed, the state monodromy jumps. b) depicts the radial quantization and is related to a) by a conformal transformation.

where $\delta^{(2)}(x)$ is Poincaré dual to $x$. Let us quantize the theory on a circle $S^1$, so that spacetime is a cylinder. The insertion of the monodromy operator corresponds to field configurations where the monodromy $\frac{1}{2\pi}\int_{S^1} d\phi$ is not conserved, i.e., it jumps as we evolve along the cylinder. In other words, if $x = (x_0, x_1) \in (S^1, \mathbb{R})$ the monodromy charge will jump at $x_1$. This is depicted in Figure 1. Field configurations with this boundary condition violate the conservation of monodromy charge.[8]

A different perspective on topological symmetries that will be useful when discussing $(-1)$-form symmetries is as follows. A theory with a topological U(1) symmetry has field configurations that can be classified by a set of integers $n_i$. In the absence of dynamical operators carrying topological charge, there is no smooth deformation of a field configuration that can change the $n_i$'s, and the path integral splits into sectors labelled by $n_i$. Denote by $\Phi$ the set of fields in a given theory. Given a field configuration we can consider which other field configurations are connected *smoothly* to it. We say that two field configurations are connected smoothly if we can deform the former to obtain the latter without passing through singular configurations or infinite action barriers.[9] This establishes an equivalence relation. A path integral that includes a particular field configuration must include any other configurations smoothly connected to it. Consider a smooth deformation parametrized by $\tau \in [0, 1]$ such that the initial field configuration is $\Phi(\tau = 0)$ and the final one is $\Phi(\tau = 1)$. The initial topological charges must match the final ones, i.e., $n_i(\tau = 0) = n_i(\tau = 1)$. If dynamical objects charged under the topological charges are included we consider their nucleation as an allowed smooth deformation and the $n_i$'s are no longer conserved. This implies that the path integral no longer splits into different sectors. It makes no sense to restrict the values of the $n_i$'s in a path integral, since they may be changed by smooth deformations.

Let us return to the case of a compact scalar on a 2d manifold $M$ and 1-cycles $\Sigma_i \in H_1(M)$.

---

[8]In some theories, zero modes of monodromy defects could be used to construct an improved current that is conserved. The existence and structure of these zero modes is UV sensitive and we will not explore it further in models of scalar fields.

[9]An example of two configurations being forced to be included by consistency arises when they are gauge equivalent. In such cases the configuration space parameter $\tau$ is circle-valued and our construction reproduces the mapping torus configurations [59]. In cases where the two configurations are not gauge equivalent but are connected smoothly our construction yields a cylinder, instead of a torus.

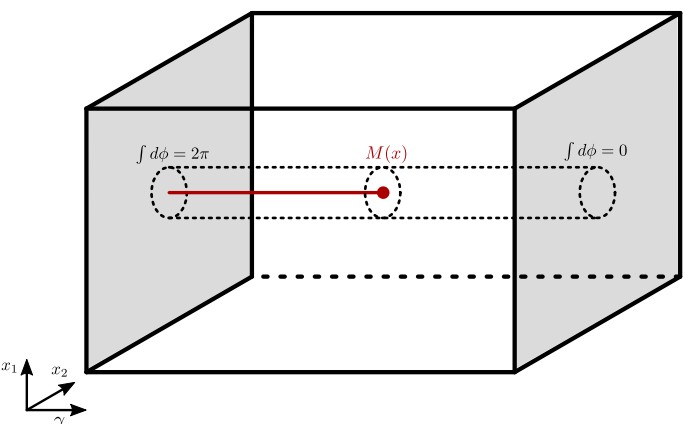

Figure 2: Smooth deformation of the field configuration in the presence of dynamical monodromy defects in a 2d compact scalar theory. The monodromy defect (line in red) ends on the monodromy defect operator $M(x)$. As a result, configurations with different monodromy charge are smoothly connected.

A field configuration has topological charges $n_i = \int_{\Sigma_i} d\phi$. If a field configuration $\phi(x)$ is smoothly deformed to $\phi'(x)$, the $n_i$ are conserved. Consider now including dynamical monodromy operators. Then, in the allowed smooth deformations we allow for insertions of these operators. Such a deformation is depicted in Figure 2. Note that in this construction the monodromy defects extend along the additional dimension and can terminate on monodromy operators. We see that the $n_i$'s are no longer conserved.

## 2.2 Two compact scalars in 2d

Consider now a theory of two compact scalars in an euclidean 2d space $M_2$ with action,

$$S = \int \frac{1}{2}|d\phi_1|^2 + \frac{1}{2}|d\phi_2|^2 + i\frac{\theta}{2\pi}d\phi_1 \wedge d\phi_2\,. \tag{4}$$

This theory has three topological symmetries with conserved currents $J_1^{(1)} = \frac{1}{2\pi}d\phi_1$, $J_2^{(1)} = \frac{1}{2\pi}d\phi_2$ and $J^{(2)} = \frac{1}{(2\pi)^2}d\phi_1 \wedge d\phi_2$. The two first currents are associated with the magnetic 0-form monodromy symmetries of the two scalar fields while the latter is a $(-1)$-form Chern-Weil symmetry for which we have turned on a $\theta$-background in Equation (4). Note that the Chern-Weil current is the product of the two magnetic currents. We will refer to symmetries arising in this way as *composite* symmetries. The two monodromy symmetries can be explicitly broken by adding dynamical monodromy operators, as in the discussion in Section 2.1. Inserting monodromy operators has a further effect: it breaks the $(-1)$-form Chern-Weil symmetry. Consider the insertion of a monodromy operator $M_1(x)$ sourcing $J_1^{(1)}$, $dJ_1^{(1)} = \delta^{(2)}(x)$. In sectors with non-trivial $\phi_2$ monodromy the Chern-Weil current is also sourced by this operator,

$$dJ^{(2)} = \frac{1}{2\pi}\delta^{(2)}(x) \wedge d\phi_2\,. \tag{5}$$

But this equation is non-sensical in 2 spacetime dimensions. This failure reflects the fact that there is no physical process that can violate a $(-1)$-form symmetry. Of course $(-1)$-form symmetries are not related with the conservation of quantities in Hilbert space. On the other hand we can still make sense of the second interpretation of topological symmetries that was presented in Section 2.1. Let us consider for definiteness $M_2$ to be a torus. There are two 1-cycles, $\Sigma_a$ and $\Sigma_b$. A field configuration will have associated topological numbers $n_i = \frac{1}{2\pi}\int_{\Sigma_i} d\phi_1$,

$m_i = \frac{1}{2\pi} \int_{\Sigma_i} \mathrm{d}\phi_2$ and $k = \frac{1}{(2\pi)^2} \int_{T^2} \mathrm{d}\phi_1 \wedge \mathrm{d}\phi_2$. Charges of composite symmetries follow from more basic charges. In this case,

$$k = n_i \Omega_{ij} m_j\,, \tag{6}$$

where $\Omega_{ij}$ is the symplectic $2 \times 2$ metric. For instance, if a field configuration has $n_i = (1, 0)$ and $m_i = (0, 1)$, it follows that $k = 1$. We can now ask which other field configurations are smoothly connected to this one, such that they must also be included in a path integral. If no monodromy operators are included the answer is clear, only field configurations with the same topological numbers. On the other hand if dynamical monodromy defects $M_1(x)$ and $M_2(x)$ sourcing the monodromy currents $J_1^{(1)}$ and $J_2^{(1)}$ are allowed, the answer changes. There are now allowed smooth deformations that arbitrarily change $n_i$ and $m_i$. This in turn also arbitrarily changes $k$, whose final value is given by $k' = n'_i \Omega_{ij} m'_j$. An example of such a deformation is presented in Figure 3.

We conclude that adding dynamical monodromy defects not only breaks the monodromy symmetries, but also the $(-1)$-form Chern-Weil symmetry. Any field configuration can now be smoothly deformed to any other and the path integral no longer factorizes into different sectors. This conclusion is hardly surprising since the $(-1)$-form Chern-Weil symmetry is *composite*. However, we will see in Sec. 4 that non-composite Chern-Weil symmetries may suffer a similar fate.

We finish this section by explaining in which sense the conservation of topological numbers follows from an extension of the non-sensical equation (5). A smooth deformation parametrized by $\tau \in [0, 1]$ yields an auxiliary 3d manifold $M_3 = M_2 \times I$, where $I$ is the interval parametrized by $\tau$. Take coordinates $(x_1, x_2, \tau)$. Consider for definiteness $\phi_1$. Extend $\phi_1(x)$ from $M_2$ to $M_3$ so that $\phi_1(x, \tau)$ interpolates between two 2d field configurations $\phi_1(x, \tau = 0) = \phi_{1,0}(x), \phi_1(x, \tau = 1) = \phi_{1,1}(x)$. Consider the 2d conservation equation on $M_2$

$$\mathrm{d}(\mathrm{d}\phi_1) = \frac{1}{2}\partial_\mu \partial_\nu \phi_1 \mathrm{d}x^\mu \wedge \mathrm{d}x^\nu = 0\,. \tag{7}$$

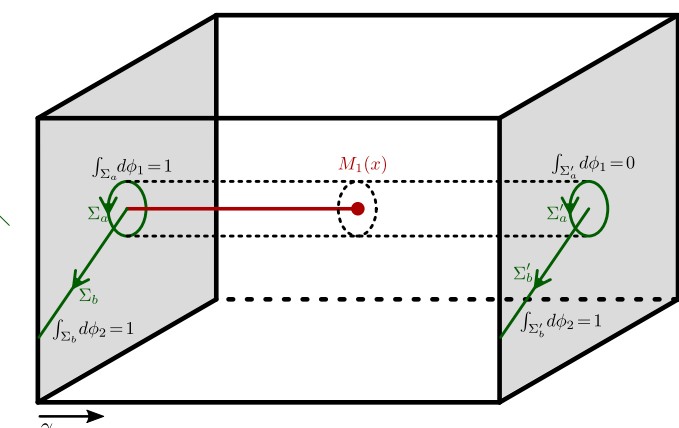

Figure 3: Smooth deformation of the field configuration in the presence of dynamical monodromy defects in a 2d theory of two compact scalars on the torus $T^2$. The physical space is shaded in grey. Its origin is removed and identified with infinity so the topology is the one of $T^2$. The fundamental cycles of $T^2$ are denoted in green. The initial field configuration has two monodromy defects that extend also in the $\gamma$ direction. We have only depicted the monodromy defect for $\phi_1$ (line in red). It ends on the monodromy defect operator $M_1(x)$. As a result, configurations with different monodromy charges are smoothly connected and $k$ jumps from $k = 1$ to $k' = 0$.

It follows that $\phi_1(x, \tau)$ obeys a similar $3d$ equation,

$$
\begin{aligned}
\mathrm{d}(\mathrm{d}\phi_1(x, \tau)) &= \frac{1}{2}\partial_a\partial_b\phi_1 \mathrm{d}x^a \wedge \mathrm{d}x^b \\
&= \mathrm{d}(\mathrm{d}\phi_1) + \partial_\mu\partial_\tau\phi_1(x, \tau)\mathrm{d}x^\mu \wedge \mathrm{d}x^\tau + \partial_\tau\partial_\mu\phi_1(x, \tau)\mathrm{d}x^\tau \wedge \mathrm{d}x^\mu \\
&= 0,
\end{aligned}
\tag{8}
$$

where we have introduced $3d$ indices $a, b$. We see that the smooth deformation satisfies $\mathrm{d}(\mathrm{d}\phi_1(x, \tau)) = 0$ and thus preserves the topological numbers $n_i$ defined above. The same holds for any Chern-Weil current, i.e., their charges are preserved under smooth deformations of the fields. In particular it also holds for $(-1)$-form Chern-Weil symmetries. This allows us to reinterpret equations such as (5). They imply that smooth deformations preserve the $(-1)$-form symmetry charge.

## 3 U(1)

In this section we will review some well-known aspects of Chern-Weil symmetries for the gauge group U(1); earlier discussions along these lines may be found in [9, 49, 60, 61]. In the case of U(1) gauge theory in $d > 4$ dimensions, it is well understood that dynamical magnetic monopoles break not only the $(d-3)$-form magnetic symmetry with current $J^{(2)} = \frac{1}{2\pi}F$ but also the $(d-5)$-form instanton number symmetry with current $J^{(4)} = \frac{1}{8\pi^2}F \wedge F$. This is straightforward: in the presence of dynamical monopoles with worldvolume $M_{d-3}$, we have $\mathrm{d}J^{(2)} = \delta_m^{(3)}[M_{d-3}]$, where $\delta_m^{(3)}[M_{d-3}]$ is the Poincaré dual of $M_{d-3}$. Then we have $\mathrm{d}J^{(4)} = \frac{1}{2\pi}F \wedge \delta_m^{(3)}[M_{d-3}]$, and similarly for the lower-form symmetries generated by higher powers of the current $F$ in higher dimensional gauge theories.

Although this derivation is simple and transparent, it is useful to give an alternative derivation of the breaking of the symmetry generated by $J^{(4)}$, in terms of whether we can couple this current to a background $(d-4)$-form gauge field $C$. This is useful for two reasons. First, it reveals that in some cases an improved symmetry current can be constructed which is conserved, using a localized dyon mode on the magnetic monopole worldvolume. Second, it generalizes to the case $d = 4$, where $\mathrm{d}J^{(4)} = 0$ holds trivially because $J^{(4)}$ is a top form.

Consider U(1) gauge theory with $J^{(4)}$ coupled to the background gauge field $C$, so that we have

$$
S = \int_{X_d} \left[ -\frac{1}{2e^2}F \wedge \star F + \frac{1}{8\pi^2}C \wedge F \wedge F \right].
\tag{9}
$$

From this theory, we can attempt to construct a magnetic dual field $A_{\mathrm{M}}$ whose field strength is

$$
\frac{1}{2\pi}F_{\mathrm{M}} = \frac{\delta\mathcal{L}^{(4)}}{\delta F} = -\frac{1}{e^2}\star F + \frac{1}{4\pi^2}C \wedge F,
\tag{10}
$$

where $\mathcal{L}^{(4)}$ is the 4-form Lagrangian density we integrate to obtain $S$. This equation can always be solved locally, since the equation of motion tells us that the exterior derivative of the right-hand side is zero. However, we see that $F_{\mathrm{M}}$ is not gauge invariant under gauge transformations of $C$. In particular,

$$
C \mapsto C + \mathrm{d}\lambda \quad \Rightarrow \quad A_{\mathrm{M}} \mapsto A_{\mathrm{M}} + \frac{1}{2\pi}\mathrm{d}\lambda \wedge A.
\tag{11}
$$

This means that we cannot define a monopole action along its worldvolume $\Gamma$ with the simple form $S_{\mathrm{M}} = \int_\Gamma (T \star_\Gamma 1 + A_{\mathrm{M}})$: it is not invariant under background $C$ gauge transformations.

Another way to say this without introducing $A_M$, in the case where the worldvolume $\Gamma$ is a boundary $\partial \Sigma = \Gamma$, is to introduce the surface operator

$$\exp\left[2\pi i \int_\Sigma \left(-\frac{1}{e^2} \star F + \frac{1}{4\pi^2} C \wedge F\right)\right].\tag{12}$$

This is equivalent to $\int_\Gamma A_M$ when it is a genuine operator on $\Gamma$ (independent of the choice of $\Sigma$). The quantization of electric flux ensures this: $\int_\Omega \left(-\frac{1}{e^2} \star F + \frac{1}{4\pi^2} C \wedge F\right) \in \mathbb{Z}$ for closed $\Omega$, so given two surfaces $\Sigma$ and $\Sigma'$ that have a shared boundary $\Gamma$, we can take $\Omega = \Sigma \cup \overline{\Sigma'}$ to see that the integrals over $\Sigma$ and $\Sigma'$ are identical (on-shell). However, (12) is not invariant under $C$ gauge transformations. One could consider a different operator obtained by integrating by parts to replace $C \wedge F$ by $A \wedge dC$, but then the operator would fail to be invariant under $A$ gauge transformations. This shows, albeit in a more cumbersome manner than simply computing $dJ^{(4)}$, that the $(d-5)$-form Chern-Weil symmetry is explicitly broken.

The worldvolume of a monopole may contain localized degrees of freedom that restore invariance under $C$ gauge transformations, which can be understood as a form of anomaly inflow [49, 62]. In some cases, like the famous 't Hooft Polyakov monopole, dyonic degrees of freedom that give electric charge to a magnetic monopole are known to exist [37, 38]. We review the argument for this in Appendix A. Suppose that our monopole has such a degree of freedom: a periodic scalar field $\sigma \cong \sigma + 2\pi$, which transform as $\sigma \mapsto \sigma - \alpha$ when $A \mapsto A + d\alpha$, so that $D\sigma \equiv d\sigma + A$ is a covariant derivative. Then we can couple $C$ to the monopole worldvolume theory according to

$$S_M = \int_\Gamma \left[T \star_\Gamma 1 + A_M - \frac{1}{2\ell^2}|D\sigma|^2 + \frac{1}{2\pi} C \wedge D\sigma\right].\tag{13}$$

The path integral includes a sum over the worldvolumes of dynamical monopoles, $\Gamma$, together with a path integral over the localized field $\sigma$ living on these worldvolumes. An 't Hooft operator can be defined in a similar manner [52], with the path integral over $\sigma$ along the operator's worldvolume being necessary for $C$ gauge invariance. We now have the requisite gauge invariance for $S_M$ under $C$ gauge transformations $C \mapsto C + d\lambda$ (in which we include the case $C \mapsto C + 2\pi\omega$ with $\omega \in H^{d-4}(X, \mathbb{Z})$, thought of as a "winding" configuration of $\lambda$):

$$\exp(iS_M) \mapsto \exp(iS_M)\exp\left[\frac{i}{2\pi} \int_\Gamma d\lambda \wedge d\sigma\right] = \exp(iS_M),\tag{14}$$

using the fact that $\frac{1}{2\pi}\int d\lambda, \frac{1}{2\pi}\int d\sigma \in \mathbb{Z}$. Thus, we infer that there actually is a $(d-5)$-form symmetry in the theory, generated by the current

$$\tilde{J}^{(4)} \equiv J^{(4)} - \frac{1}{2\pi}(d\sigma + A) \wedge \delta_m^{(3)}[M_{d-3}],\tag{15}$$

for which $d\tilde{J}^{(4)} = 0$ follows directly from $dJ^{(2)} = \delta_m^{(3)}[M_{d-3}]$ and $d(d\sigma + A) = F$.

The case $d = 4$ proceeds in an exactly analogous manner: $C$ is now viewed as a background axion field $\theta$, and we require invariance under its large gauge transformations $\theta \cong \theta + 2\pi$ (we can phrase this as $J^{(4)}$ generating a U(1) $(-1)$-form symmetry [30]). The monopole obstructs the coupling of the theory to $\theta$ in precisely the same manner as in $d > 4$, but if the monopole admits a dyon mode $\sigma$, we can construct an improved current which takes precisely the same form (15). From the viewpoint taken in Sec. 2, the operator $\exp[i\alpha\tilde{J}^{(4)}]$ is then topological in 5d configuration space.

# 4 PSU($n$) Chern-Weil breaking by monopoles

## 4.1 The main idea

PSU($n$) Yang-Mills theory in $d$ dimensions has a $\mathbb{Z}_n$ magnetic $(d-3)$-form symmetry, and a U(1) instantonic $(d-5)$-form symmetry. On a spacetime $M$, the magnetic symmetry current is given by the $\mathbb{Z}_n$-valued Stiefel-Whitney class $w_2 \in H^2(M, \mathbb{Z}_n)$, which measures the obstruction of lifting a PSU($n$) bundle to an SU($n$) bundle. This class is normalized so that the integral $\int_\Sigma w_2$ over a closed 2-cycle $\Sigma$ takes values in the integers mod $n$, which we can take to be $\{0, 1, \ldots, n-1\}$. The instanton number of a PSU($n$) gauge bundle, given by the integral $\frac{1}{8\pi^2} \int_X \operatorname{tr}(F \wedge F)$ over a closed 4-cycle $X$, is an integer multiple of the base unit $1/n$ (whereas for SU($n$) it would be an integer). Furthermore, the instanton number can only be fractional for PSU($n$) bundles that are not SU($n$) bundles. More specifically, the instanton number $I$ integrated over a closed 4-manifold $X$ is related to the Stiefel-Whitney class $w_2$ as follows:

$$I \equiv \frac{1}{8\pi^2} \int_X \operatorname{tr}(F \wedge F) = \frac{n-1}{n} \int_X \frac{w_2 \wedge w_2}{2} \quad \mod 1 \,. \tag{16}$$

We will focus on the case of $n$ odd to avoid subtleties in the definition of the Pontryagin square. See, e.g., [5, 29, 36] for further discussion.

The magnetic $(d-3)$-form symmetry is explicitly broken if dynamical magnetic monopoles exist. If $j_{\mathrm{mag}}$ is a magnetic monopole current, normalized so that the minimal magnetic monopole has charge 1, then we have an equation of the form

$$\mathrm{d}w_2 = j_{\mathrm{mag}} \,. \tag{17}$$

Both sides of this equation are valued in $H^3(M, \mathbb{Z}_n)$. Here, we see that $w_2$ plays a role analogous to $\frac{1}{2\pi} F$ in U(1) gauge theory.

Much as in the U(1) case, the equations (16) and (17) imply that, in $d > 4$ dimensions, instanton number is not conserved when magnetic monopoles are dynamical objects in the theory:

$$\Delta I = \frac{1}{8\pi^2} \int_X \mathrm{d}\left[\operatorname{tr}(F \wedge F)\right] = \frac{n-1}{n} \int_X w_2 \wedge \mathrm{d}w_2 = \frac{n-1}{n} \int_X w_2 \wedge j_{\mathrm{mag}} \quad \mod 1 \,. \tag{18}$$

In particular, this shows that instanton number will be violated by fractional amounts (multiples of $1/n$) when we have a magnetic monopole current $j_{\mathrm{mag}}$ *and* a fractional magnetic flux $w_2$ on a separate 2-cycle. This, in turn, implies that the existence of a minimally charged magnetic monopole fully breaks the instantonic symmetry in PSU($n$) gauge theory.

This argument generalizes immediately to other gauge groups of the form $G/\Gamma$ for simply connected $G$ and nontrivial $\Gamma \subset Z(G)$, for which the instanton number of $G/\Gamma$ can be a fraction of the allowed instanton number for $G$; see Table 1 of [29].

In the discussion below, we will see how this works more explicitly by directly studying field configurations with nontrivial magnetic flux and instanton number, and seeing how the insertion of a monopole annihilation operator can change any initial instanton number to zero. We will also examine the 4d case, where breaking of the instantonic symmetry corresponds to an inability to couple magnetic monopoles to a background axion field.

## 4.2 Example field configuration with fractional instanton number

We exhibit a PSU($n$) field configuration on the space $S^2 \times S^2$ that has magnetic flux on each sphere, and instanton number on the full 4d space. There is a well-known construction of

a Dirac monopole using two coordinate charts, one covering the northern part of the sphere ($\theta \neq \pi$) and one covering the southern part ($\theta \neq 0$). Analogously, we cover $S^2 \times S^2$ with four coordinate charts, which we label $(++)$, $(+-)$, $(-+)$, and $(--)$, where the first $+$ or $-$ refers to whether we cover the northern or southern part of the first $S^2$ factor respectively, and similarly for the second $+$ or $-$. We take our gauge field configuration to be

$$A_{(\sigma_1\sigma_2)} = \frac{1}{2}T_1(\sigma_1 1 - \cos\theta_1)\mathrm{d}\phi_1 + \frac{1}{2}T_2(\sigma_2 1 - \cos\theta_2)\mathrm{d}\phi_2\,, \tag{19}$$

where $\sigma_1, \sigma_2 \in \{+,-\}$ are signs and $T_1$, $T_2$ are matrices in $\mathfrak{su}(n)$ for which $\exp(2\pi\mathrm{i}T_k) \in \mathbb{Z}_n \subset \mathrm{SU}(n)$. If $\exp(2\pi\mathrm{i}T_k)$ is not the identity, then we have a field configuration that is not an $\mathrm{SU}(n)$ bundle but is a $\mathrm{PSU}(n)$ bundle due to the $\mathbb{Z}_n$ quotient defining $\mathrm{PSU}(n)$. In these cases, the field configuration has nontrivial Stiefel-Whitney invariants.

In order to have a valid gauge field configuration, we need our field configurations to be related by gauge transformations $g_{U \to V}$ on the overlap of two coordinate charts $U \cap V$, i.e.,

$$A_V = A_U - \mathrm{i}g_{U \to V}^{-1}\mathrm{d}g_{U \to V}\,. \tag{20}$$

For consistency, we further require that $g_{V \to U} = g_{U \to V}^{-1}$ and that the cocycle condition $g_{V \to W}(x) \cdot g_{U \to V}(x) = g_{U \to W}(x)$ is satisfied on triple overlaps $U \cap V \cap W$. For our case, we take:

$$\begin{aligned}
g_{(++) \to (-+)} &= g_{(+-) \to (--)} = \exp(-\mathrm{i}T_1\phi_1)\,, \\
g_{(++) \to (+-)} &= g_{(-+) \to (--)} = \exp(-\mathrm{i}T_2\phi_2)\,, \\
g_{(++) \to (--)} &= \exp(-\mathrm{i}T_1\phi_1)\exp(-\mathrm{i}T_2\phi_2)\,, \\
g_{(+-) \to (-+)} &= \exp(-\mathrm{i}T_1\phi_1)\exp(+\mathrm{i}T_2\phi_2)\,.
\end{aligned} \tag{21}$$

These choices are obviously consistent with the cocycle conditions provided that

$$[T_1, T_2] = 0\,. \tag{22}$$

The gauge field strength of this configuration is

$$F = \mathrm{d}A - \mathrm{i}A \wedge A = \frac{1}{2}T_1\mathrm{vol}_{S_1^2} + \frac{1}{2}T_2\mathrm{vol}_{S_2^2}\,, \tag{23}$$

and the instanton number is

$$I = \frac{1}{8\pi^2}\int \mathrm{tr}(F \wedge F) = \mathrm{tr}(T_1 T_2)\,. \tag{24}$$

As noted above, a nontrivial value of $\exp(2\pi\mathrm{i}T_k)$ indicates that the field configuration is not an $\mathrm{SU}(n)$ bundle. Specifically, if $\exp(2\pi\mathrm{i}T_k) = \exp(2\pi\mathrm{i}z_k/n)$, with $z_k \in \{0, 1, \ldots n-1\}$, then we can write the Stiefel-Whitney class in $H^2(S^2 \times S^2, \mathbb{Z}_n)$:

$$w_2 = z_1\frac{\mathrm{vol}_{S_1^2}}{4\pi} + z_2\frac{\mathrm{vol}_{S_2^2}}{4\pi}\,. \tag{25}$$

As a specific example, consider the choice

$$T_1 = \frac{j_1}{n}\mathrm{diag}(1, 1, \ldots, 1, 1-n)\,, \qquad T_2 = \frac{j_2}{n}\mathrm{diag}(1-n, 1, \ldots, 1, 1)\,, \tag{26}$$

with $j_1, j_2 \in \mathbb{Z}$. This gives

$$I = -\frac{1}{n}j_1 j_2\,. \tag{27}$$

As expected, then, we can obtain fractional $\text{PSU}(n)$ instanton number, but to do so we require field configurations where $j_1$ and $j_2$ are both not divisible by $n$—that is, field configurations with nontrivial Stiefel-Whitney class. Specifically, we have Stiefel-Whitney class

$$w_2 = j_1 \frac{\text{vol}_{S_1^2}}{4\pi} + j_2 \frac{\text{vol}_{S_2^2}}{4\pi} \, , \tag{28}$$

and

$$\int_{S^2 \times S^2} \frac{1}{2} w_2 \wedge w_2 = j_1 j_2 \mod n \, . \tag{29}$$

This reflects the general relationship (16).

## 4.3 Destruction of instanton number with a monopole operator

Let us now consider a field configuration on $\mathbb{R}^{3,1} \times S^2$, which at time $t < 0$ is given by the field configuration (19), but now with the first $S^2$ factor viewed as sitting inside $\mathbb{R}^3$. In other words, we have a magnetic monopole with charge $j_1/n$ inside $\mathbb{R}^3$, and a magnetic flux $j_2/n$ on a separate $S^2$ factor. For clarity, we will specialize to the case $j_1 = 1$, so that the monopole has minimal (fractional) charge.

This is a six dimensional theory, so a magnetic monopole has a three-dimensional worldvolume and is created or annihilated by a 2-dimensional surface operator.

At time $t = 0$, we can insert such a monopole annihilation operator to destroy the flux on the first $S^2$. Specifically, we insert this operator at the origin of $\mathbb{R}^3$, and wrapping the second $S^2$. At time $t > 0$, we have a trivial field configuration on $\mathbb{R}^3$, while a nontrivial configuration of magnetic flux $j_2/n$ exists on the second $S^2$. This final field configuration has instanton number zero, and Stiefel-Whitney class $w_2 = j_2 \frac{\text{vol}_{S_2^2}}{4\pi}$.

The integer $j_2$ could take any value, and so we see that inserting a monopole annihilation operator of minimal (fractional) charge can destroy *any* initial instanton number, even an integer value.

### 4.3.1 Brief aside on unstable monopoles

Let us digress to address a possible confusion one may have. Suppose that we had instead considered larger $j_1$, say, $j_1 = n + 1$. In this case, the monopole field configuration on $\mathbb{R}^{3,1}$ is unstable. The conserved monopole charge lies in $\mathbb{Z}_n$, so by emitting gluons, a monopole with $j_1 = n + 1$ will decay to a monopole of charge $j_1 = 1$ [63,64]. It might seem that this process could change the instanton number (27), without requiring insertion of a monopole creation or annihilation operator. This is not the case. The problem with this logic is that the field configuration on $\mathbb{R}^{3,1}$ is not independent of that on $S^2$: a homotopy that takes $j_1 = n + 1$ to $j_1 = 1$ necessarily passes through field configurations that change the first term in (19) so that it no longer commutes with $T_2$, and hence the cocycle conditions can't be satisfied without also changing the field configuration on $S^2$. Tracking the time evolution of such an unstable monopole configuration requires following the full field configuration on both the $\mathbb{R}^{3,1}$ factor and the $S^2$ factor consistently, and is a nontrivial task. In any case, topological invariance assures us that the instanton number $I$ will remain unchanged, provided we do not insert monopole creation or annihilation operators.

## 4.4 The PSU($n$) Witten effect

The previous discussion applies to spacetime dimensions $d \geq 4$. The case $d = 4$ is special for various reasons. Firstly, the Chern-Weil symmetry is $(-1)$-form. Secondly, the four dimensional

Yang-Mills theory admits a $\theta$-term that switches on a non-trivial background for the $(-1)$-form symmetry. In this subsection we consider the presence of 't Hooft lines with a $\theta$-term and derive the PSU$(n)$ Witten effect.[10] In section 4.4.1 we show that promoting $\theta$ to a dynamical field requires the addition of dyonic modes on the monopoles.

Let us consider PSU$(n)$ gauge theory in 4 spacetime dimensions. In manifolds with non-trivial 2-cycles, gauge bundles are labeled by the second Chern class and $w_2 \in H^2(M, \mathbb{Z}_n)$, the obstruction to lifting the PSU$(n)$ bundle to a SU$(n)$ one. We can describe the sum over $w_2$ starting with an SU$(n)$ path integral and a $B_e \in H^2(M, \mathbb{Z}_n)$ background for the electric $\mathbb{Z}_n^{(1)}$ symmetry. We further introduce a dual Lagrange multiplier field $\tilde{B} \in H^2(M, \mathbb{Z}_n)$ that identifies $w_2$ with $B_e$. In practice, we add the following term to the action[11]

$$S \supset \frac{2\pi i}{n} \int \tilde{B} \wedge (w_2 - B_e).$$ (30)

Summing over $B_e$ gauges the electric symmetry[12] and promotes the SU$(n)$ path integral to PSU$(n)$. The PSU$(n)$ theory has a $\mathbb{Z}_n^{(1)}$ magnetic symmetry with symmetry operators,

$$U_q[\Sigma_2] = e^{2\pi i \frac{q}{n} \oint_{\Sigma_2} B_e}.$$ (31)

Consider now carrying out the sum over $w_2$. It enforces a quantization condition that yields the following operator equation

$$e^{\frac{2\pi i}{n} \oint \tilde{B}} = 1.$$ (32)

This equation implies that we can define the following genuine line operators

$$H_p[\gamma] = e^{\frac{2\pi i p}{n} \int_D \tilde{B}},$$ (33)

where $D$ is a disk whose boundary is a closed curve $\gamma$. It is easy to check that the following operator equation holds[13]

$$U_q[\Sigma_2] H_p[\gamma] = e^{\frac{2\pi i p q \cdot \text{Link}(\gamma, \Sigma_2)}{N}} H_p[\gamma].$$ (34)

This equation implies that $H_p[\gamma]$ are 't Hooft lines for the PSU$(n)$ monopoles. Consider now adding a $\theta$-term, which in PSU$(n)$ includes the fractional piece in eq. (16). The $w_2$-dependent terms in the action are

$$S \supset \frac{2\pi i}{n} \int \tilde{B} \wedge w_2 + i\theta \frac{n-1}{n} \int \frac{w_2 \wedge w_2}{2}.$$ (35)

Performing the sum over $w_2$ now yields a modified quantization condition that implies the following operator identity

$$e^{i\left(\oint \frac{2\pi}{n} \tilde{B} + \theta \frac{n-1}{n} w_2\right)} = 1.$$ (36)

It follows that the correct genuine 't Hooft lines are now[14]

$$H'_p[\gamma] = e^{\frac{ip}{n} \int_D \left(2\pi \tilde{B} + \theta(n-1) w_2\right)}.$$ (37)

By virtue of eq. (36) this 't Hooft line does not depend on the choice of $D$ as long as $\partial D = \gamma$. We see that the monopole is dressed with $w_2$, which plays the role of $F$ in the U(1) gauge theory. This is the PSU$(n)$ version of the Witten effect.

---

[10]We thank Iñaki García-Etxebarria for discussions and feedback on this section.

[11]We choose to use the (slightly imprecise) notation of differential forms even if $w_2, B_e$ and $\tilde{B}$ are cocycles to avoid introducing mathematical machinery not used in the rest of the text.

[12]Note that in this description of the SU$(n)$ theory we are already summing over $w_2$.

[13]A field redefinition $\tilde{B} \to \tilde{B} + q\delta^{(2)}[\Sigma^2]$ removes the symmetry operator and leaves behind the desired phase.

[14]Note that eqs. (36) and (37) are not gauge invariant under $w_2$ gauge transformations $w_2 \to w_2 + n\Lambda_2$, $\Lambda_2 \in H^2(M, \mathbb{Z})$. Gauge invariant expressions involve a dressing by the PSU$(N)$ field strength [1, 2]. We stick to eqs. (36) and (37), which are enough for our purposes.

### 4.4.1 Dyon modes

It is apparent from eq. (37) that promoting the $\theta$-angle to a dynamical axion field renders the 't Hooft lines non-gauge invariant. This is similar to the discussion in section 3 in the U(1) gauge theory. In that case, gauge invariant 't Hooft lines can exist, provided that they are dressed with the correct dyonic modes. As we now discuss, the same is true in the case at hand, as long as the lines in eq. (37) are dressed with a dyonic compact scalar field.

A discrete $\mathbb{Z}_n$ gauge field, or a characteristic class, can be understood as a U(1) field strength $F$ subject to the following gauge invariance:

$$F \to F + n\Lambda_2, \qquad \Lambda_2 \in H^2(X, \mathbb{Z}). \tag{38}$$

It follows that $\oint \frac{1}{2\pi} F = 0, ..., n-1$ and $\oint A = 0$, where locally $F = \mathrm{d}A$. That is, $\frac{1}{2\pi} F$ is a $\mathbb{Z}_n$ 2-form gauge field. We use this construction for $w_2$. Since it is closed, it may be locally expressed as $w_2 = \frac{1}{2\pi} \mathrm{d}A_e$, subject to the following gauge invariances,

$$A_e \to A_e + n\lambda_1 + \mathrm{d}\lambda_0, \qquad \oint \mathrm{d}\lambda_1, \qquad \oint \mathrm{d}\lambda_0 \in 2\pi\mathbb{Z}. \tag{39}$$

It follows that $\oint A_e = 0$ and $\oint \frac{1}{2\pi} \mathrm{d}A_e = 0, ..., n-1$. Consider now a compact scalar field $\mathrm{d}\sigma$ transforming under the $A_e$ gauge transformation as,

$$\sigma \to \sigma - \lambda_0. \tag{40}$$

This field can be used to write a gauge invariant 't Hooft line,[15] even after $\theta$ is promoted to a dynamical field,

$$\tilde{H}'_p[\gamma] = H'_p[\gamma] e^{\frac{-ip(n-1)}{n} \int_\gamma \frac{\theta}{2\pi}(A_e - \mathrm{d}\sigma)}. \tag{41}$$

We learn that a dyonic mode is needed to define gauge invariant 't Hooft lines in the PSU($n$) theory once $\theta$ is made dynamical. If $\theta$ is not dynamical, the dyonic mode compensates the spectral flow as $\theta \to \theta + 2\pi$. As in the U(1) case, the dyonic modes may or may not be present, depending on the concrete UV theory under consideration.

## 4.5 The improved PSU($n$) Chern-Weil current

In U(1) gauge theory we saw that dynamical monopoles break the Chern-Weil symmetry. But that was not the end of the story. The dyonic mode living on the monopole worldline came with the correct couplings such that an improved, conserved Chern-Weil current could be defined. As we will see this holds for PSU($n$) as well.

Consider the non-conservation of the instanton number in eq. (18). In the $4d$ case this equation should be extended to the auxiliary manifold $X_5 = X_4 \times I$, where $X_4$ is the spacetime manifold and $I$ the interval in configuration space. If the smooth deformation of the fields includes an insertion of a dynamical monopole sourcing a magnetic current $j_{\mathrm{mag}}$, the instanton number is not conserved

$$\Delta I = \frac{n-1}{n} \int_{X_5} w_2 \wedge j_{\mathrm{mag}} \mod 1, \tag{42}$$

where $j_{\mathrm{mag}} = \delta^{(3)}[\gamma]$. We can use the dyonic modes on the monopole worldline in eq. (37) to write an improved instanton number. Its fractional part is given by

$$\tilde{I} = \frac{n-1}{n} \int_{X_4} \left[ \frac{w_2 \wedge w_2}{2} - \frac{1}{2\pi}(A_e - \mathrm{d}\sigma) \wedge j_{\mathrm{mag}} \right] \mod 1, \tag{43}$$

---

[15]Amusingly, eq. (41) is a well defined gauge invariant operator, even if eq. (37) is not.

which is conserved under the smooth deformation considered above,

$$\Delta \tilde{I} = 0 \,. \tag{44}$$

Here we do not give a full expression for $\tilde{I}$ that defines the integer part as well as the fractional part, which would require addressing the subtlety mentioned in footnote 14. However, in various explicit examples in Sec. 5 below we will obtain such expressions (and see that they can be dependent on the orientation of the magnetic monopole charge within the gauge group).

## 5 Chern-Weil symmetries, Higgsing, and dyons

We have so far been mostly agnostic about the UV nature of the monopoles and just assumed that they exist and have a finite energy. In this section we explore explicit examples of theories with finite energy monopole configurations, specifically, non-abelian gauge theories with gauge group $G$ Higgsed down to a subgroup $H$. We provide 4 examples of adjoint Higgsing, including the SU(5) GUT theory. In these examples the UV Chern-Weil instanton number current is exactly matched in the IR once the breaking by the monopoles is taken into account. As in the U(1) and PSU($n$) cases explored in Sec. 3 and Sec. 4, the dyon mode on the monopole is essential for this matching. After these examples, we step back and examine the general conditions that are needed for monopoles and dyons in the IR EFT to preserve precisely the UV instanton number symmetry. We will see that these conditions are not automatic, and may no longer hold in examples where the Higgs field is in a general representation. We close the section by exploring such examples, for some of which we find that there are emergent Chern-Weil symmetries in the IR that are not explicitly broken by magnetic monopoles.

### 5.1 SU(2) → U(1)

The best-known example is the 't Hooft-Polyakov monopole, which arises in the adjoint Higgsing of SU(2) gauge theory down to U(1). In the UV, the theory has a conserved[16] Chern-Weil current,

$$J_{\text{UV}} = \frac{1}{8\pi^2} \text{tr}(F \wedge F) \,, \tag{45}$$

where the SU(2) gauge fields are packaged into a matrix[17]

$$A = A_i \tau^i = \frac{1}{2} A_i \sigma^i \,, \tag{46}$$

with field strength $F = \mathrm{d}A$. The current is normalized such that its associated charge (the instanton number) takes any value in $\mathbb{Z}$.

We fix a gauge in which the unbroken U(1) generator is proportional to $A_3$. In this normalization, however, the minimum charge is $1/2$. If we fix a conventional normalization, so that the minimum electric charge is 1, we have the U(1) generator

$$a = \frac{1}{2} A_3 \,, \tag{47}$$

with field strength $f = \mathrm{d}a$. In other words, $\sigma^3$ is normalized nicely for a U(1) generator, in the sense that

$$\exp(2\pi i \sigma^3) = \mathbb{1} \,. \tag{48}$$

---

[16]This has the standard meaning that $\mathrm{d}J_{\text{UV}} = 0$ for $d > 4$, and means that the current can be coupled to a background axion field in $d = 4$.

[17]Here $\sigma_i$ are the Pauli matrices.

The 't Hooft-Polyakov monopole has minimal flux

$$\frac{1}{2\pi} \oint f = \frac{1}{4\pi} \oint F_3 = 1 \,. \tag{49}$$

Correspondingly, we have,

$$\mathrm{d}\left(\frac{1}{2\pi} f\right) = j_{\mathrm{mag}} \,, \tag{50}$$

where $j_{\mathrm{mag}}$ is the monopole current. This current is the Poincaré dual to the monopole worldline $\gamma$, so we denote it $j_{\mathrm{mag}} = \delta^{(3)}[\gamma]$.

The UV Chern-Weil current relates to the unbroken U(1) gauge field via

$$J_{\mathrm{UV}} = \frac{1}{16\pi^2} F_3 \wedge F_3 + \cdots = \frac{1}{4\pi^2} f \wedge f + \cdots \,, \tag{51}$$

where $\cdots$ are terms involving the massive gauge fields $A_{1,2}$. In U(1) gauge theory, we expect that the Chern-Weil current normalized to have integer periods is

$$j = \frac{1}{8\pi^2} f \wedge f \,, \tag{52}$$

so the term in $J$ is twice as large. (This factor of 2 was discussed from a rather different viewpoint recently in [65], but we believe the viewpoints should ultimately be equivalent once one properly justifies the boundary conditions assumed in that paper.)

In the presence of an 't Hooft-Polyakov monopole the naive IR Chern-Weil current $j$ is no longer conserved,

$$\mathrm{d}j = \frac{1}{4\pi^2} f \wedge \mathrm{d}f = \frac{1}{2\pi} f \wedge \delta^{(3)}[\gamma] \,. \tag{53}$$

(In the case $d = 4$, we can interpret the exterior derivative here in terms of the extension to configuration space discussed in Sec. 2, or we can reintrepret this result in terms of the Witten effect obstructing a background axion's coupling to $j$.) However, there is an improved current due to the dyon mode on the monopole worldline, as we will show below.

Where does the dyon mode come from? The core of the 't Hooft-Polyakov monopole has a nontrivial classical solution for the $W$ bosons, i.e., the gauge fields $A_{1,2}$. There is a classical zero mode of the solution which simply rephases these by a U(1) transformation. This zero mode corresponds to a mode on the monopole worldline which is a compact boson $\sigma \cong \sigma + 2\pi$ that has the same charge as a $W$ boson, which is to say, charge 1 under $A_3$ or, equivalently, charge 2 under $a$. This means that $\sigma$ appears via a covariant derivative[18]

$$\mathrm{D}\sigma \equiv \mathrm{d}\sigma - 2a \,. \tag{54}$$

Then, the 4$d$ monopole action will contain a term whose exterior derivative is

$$\mathrm{d}\left(\frac{1}{2\pi} \mathrm{D}\sigma \wedge \delta_{\mathrm{M}}\right) = -\frac{1}{\pi} f \wedge \delta_{\mathrm{M}} \,. \tag{55}$$

The factor of $\frac{1}{2\pi}$ is chosen to be the correct normalization for a minimally winding configuration of $\sigma$ to carry charge 1 under this current. This shows that we can have a *conserved* Chern-Weil current in the IR that takes the form

$$J_{\mathrm{IR}} = \frac{1}{4\pi^2} f \wedge f + \frac{1}{2\pi} \mathrm{D}\sigma \wedge \delta_{\mathrm{M}} \,. \tag{56}$$

This is the IR current that matches the UV current $J$; the normalization of the $f \wedge f$ term is exactly as expected (and *not* what one might have naively thought from the IR viewpoint).

---

[18]See Appendix A for more details on the dyon action and its derivation.

## 5.2  SU(3) → U(1) × U(1)

A more interesting example involves breaking SU(3) to the maximal torus U(1) × U(1) with a generic adjoint VEV. The UV theory still has a single Chern-Weil current, $J_{\text{UV}} = \frac{1}{8\pi^2}\text{tr}(F \wedge F)$, while there are now three different IR Chern-Weil currents to consider, of the form $f_i \wedge f_j$. We expect these currents to be no longer conserved in the presence of dynamical monopoles, but it may be possible to define an improved $J_{\text{IR}}$ current which is still conserved. We will see that the dyonic mode action is such that $J_{\text{IR}} = J_{\text{UV}}$.

The generators of the unbroken U(1) factors can be taken to be the Gell-Mann matrices

$$\lambda^3 = \begin{pmatrix} 1 & 0 & 0 \\ 0 & -1 & 0 \\ 0 & 0 & 0 \end{pmatrix}, \qquad \lambda^8 = \frac{1}{\sqrt{3}} \begin{pmatrix} 1 & 0 & 0 \\ 0 & 1 & 0 \\ 0 & 0 & -2 \end{pmatrix}. \tag{57}$$

The Gell-Mann matrices are chosen such that $\text{tr}(\lambda^i \lambda^j) = 2\delta^{ij}$. This is convenient, but also sometimes awkward from the viewpoint of the normalization of U(1) subgroups. For example, we have

$$\exp(2\pi i\lambda^3) = \mathbb{1}, \qquad \exp(2\pi i\sqrt{3}\lambda^8) = \mathbb{1}. \tag{58}$$

One might then be tempted to view $\lambda^3$ and $\sqrt{3}\lambda^8$ as the generators of two independent U(1) groups, determining the charge lattice of the IR theory. However, this is not quite right. We should not think of the unbroken group as a product of the U(1) generated by $\lambda^3$ and that generated by $\lambda^8$, because there is a nontrivial relation

$$\exp(-i\pi\lambda^3)\exp(i\pi\sqrt{3}\lambda^8) = \mathbb{1}. \tag{59}$$

Instead, consider the generator of the lower-right embedding of SU(2),

$$\lambda'^3 = \begin{pmatrix} 0 & 0 & 0 \\ 0 & 1 & 0 \\ 0 & 0 & -1 \end{pmatrix} = -\frac{1}{2}\lambda^3 + \frac{\sqrt{3}}{2}\lambda^8. \tag{60}$$

This clearly generates a U(1) ⊂ U(1) × U(1) ⊂ SU(3) and is independent of the U(1) generated by $\lambda^3$.

Writing the gauge fields as

$$A = A_i T^i = \frac{1}{2}A_i \lambda^i, \tag{61}$$

we see that the UV Chern-Weil current is

$$J = \frac{1}{8\pi^2}\text{tr}(F \wedge F) = \frac{1}{16\pi^2}\left[F_3 \wedge F_3 + F_8 \wedge F_8 + \cdots\right], \tag{62}$$

where the $\cdots$ depend on the other $A_i$ that are not neutral under U(1) × U(1) and become massive. Expressed in term of the SU(3) gauge field, the most general IR current $j$ takes the form

$$j = \frac{1}{16\pi^2}\left[\alpha F_3 \wedge F_3 + \beta F_8 \wedge F_8 + 2\gamma F_3 \wedge F_8\right]. \tag{63}$$

In the previous section there was a single monopole to consider, but there are two magnetic charges now, generating a whole lattice of monopoles. Our task is to find a Chern-Weil current that is conserved under an insertion of any such monopole. First, let's consider an 't Hooft-Polyakov monopole M embedded in the upper-left SU(2), generated by

$$\tau^i = \begin{pmatrix} \sigma^i & 0 \\ 0 & 0 \end{pmatrix} = \lambda^i, \quad i \in \{1, 2, 3\}. \tag{64}$$

It carries magnetic charge proportional to $\lambda^3$. In terms of the IR theory, we have fluxes on a sphere $\Sigma$ surrounding this monopole given by

$$\int_\Sigma \frac{1}{4\pi} F_3 = 1\,, \qquad \int_\Sigma \frac{1}{4\pi} F_8 = 0\,. \tag{65}$$

In particular this means that $dF_3 = 4\pi\delta_M$. The $W$ bosons in the core of this monopole carry charge under $\lambda^3$ and are neutral under $\lambda^8$, because $[\lambda^3, \lambda^8] = 0$. The dyon excitation rephases the $W$ bosons, and the dyon mode $\sigma$ carries charge 1 under $A_3$:

$$\begin{aligned} D\sigma &= d\sigma - A_3\,, \\ d(D\sigma) &= -F_3\,. \end{aligned} \tag{66}$$

This allows us to construct an improved current using an integer multiple of the properly quantized term $\frac{1}{2\pi} D\sigma \wedge \delta_M$. The exterior derivative of this term is,

$$d\left(\frac{1}{2\pi} D\sigma \wedge \delta_M\right) = -\frac{1}{2\pi} F_3 \wedge \delta_M\,. \tag{67}$$

Thus, in the presence of M, the naive current in eq. (63) is no longer conserved,

$$dj = \frac{1}{2\pi}\alpha F_3 \wedge \delta_M + \frac{1}{2\pi}\gamma F_8 \wedge \delta_M\,. \tag{68}$$

The first term can be canceled by a contribution of the form (67) if $\alpha \in \mathbb{Z}$, but the second term cannot, and we conclude that $\gamma = 0$ in any conserved Chern-Weil current. So far, $\beta$ is undetermined.

Next, consider a *different* 't Hooft-Polyakov monopole M$'$ from the embedding in the lower-right SU(2), generated by

$$\tau'^i = \begin{pmatrix} 0 & 0 \\ 0 & \sigma^i \end{pmatrix}\,, \quad i \in \{1, 2, 3\}\,. \tag{69}$$

In terms of the Gell-Mann matrices, $\tau'^1 = \lambda^6$ and $\tau'^2 = \lambda^7$, while $\tau'^3$ is the matrix we previously introduced as $\lambda'^3$. By analogy to the case above, we can define the orthogonal generator

$$\lambda'^8 = \frac{1}{\sqrt{3}} \begin{pmatrix} -2 & 0 & 0 \\ 0 & 1 & 0 \\ 0 & 0 & 1 \end{pmatrix}\,. \tag{70}$$

Then we can change basis according to

$$A_3 \lambda^3 + A_8 \lambda_8 = A'_3 \lambda'^3 + A'_8 \lambda'^8\,, \tag{71}$$

where multiplying both sides by generators and taking traces allows us to conclude that

$$A_3 = -\frac{1}{2} A'_3 - \frac{\sqrt{3}}{2} A'_8\,, \tag{72}$$

$$A_8 = \frac{\sqrt{3}}{2} A'_3 - \frac{1}{2} A'_8\,. \tag{73}$$

If we surround M$'$ by a sphere $\Sigma'$ we find fluxes

$$\int_{\Sigma'} \frac{1}{4\pi} F'_3 = 1\,, \qquad \int_{\Sigma'} F'_8 = 0\,, \tag{74}$$

and hence

$$\int_{\Sigma'} \frac{1}{4\pi} F_3 = -\frac{1}{2}, \qquad \int_{\Sigma'} \frac{1}{4\pi} F_8 = \frac{\sqrt{3}}{2}. \tag{75}$$

The dyon mode, corresponding to rephasing the $W$ bosons in the core of the monopole, has charge 1 under $A_3'$ and 0 under $A_8'$, so it has covariant derivative,

$$\mathrm{D}\sigma' = \mathrm{d}\sigma' - A_3', \qquad \mathrm{d}(\mathrm{D}\sigma') = -F_3' = \frac{1}{2}F_3 - \frac{\sqrt{3}}{2}F_8. \tag{76}$$

Thus we can consider currents that include a term $\frac{1}{2\pi}\mathrm{D}\sigma' \wedge \delta_{\mathrm{M}'}$, which has

$$\mathrm{d}\left(\frac{1}{2\pi}\mathrm{D}\sigma' \wedge \delta_{\mathrm{M}'}\right) = -\frac{1}{4\pi}F_3 \wedge \delta_{\mathrm{M}'} + \frac{\sqrt{3}}{4\pi}F_8 \wedge \delta_{\mathrm{M}'}. \tag{77}$$

Now, in the presence of this monopole, and taking into account that $\gamma = 0$, the naive IR Chern-Weil current is not conserved as,

$$\begin{aligned}
\mathrm{d}\left(\frac{1}{16\pi^2}\left[\alpha F_3 \wedge F_3 + \beta F_8 \wedge F_8\right]\right) &= \frac{1}{8\pi^2}\left(\alpha F_3 \wedge \mathrm{d}F_3 + \beta F_8 \wedge \mathrm{d}F_8\right) \\
&= \frac{1}{2\pi}\left[\alpha F_3 \wedge \delta_{\mathrm{M}} + \left(-\frac{\alpha}{2}F_3 \wedge \delta_{\mathrm{M}'} + \frac{\sqrt{3}\beta}{2}F_8 \wedge \delta_{\mathrm{M}'}\right)\right].
\end{aligned} \tag{78}$$

Comparing to (77), we see that we can improve this current using the localized dyon terms only in the case $\beta = \alpha$. One could also consider additional monopoles, but they would not provide new information. We conclude that the unique Chern-Weil current that is conserved in the presence of any monopole and has integer quantized charges is,

$$J_{\mathrm{IR}} = \frac{1}{16\pi^2}\left(F_3 \wedge F_3 + F_8 \wedge F_8\right) + \frac{1}{2\pi}\left(\mathrm{D}\sigma \wedge \delta_{\mathrm{M}} + \mathrm{D}\sigma' \wedge \delta_{\mathrm{M}'}\right), \tag{79}$$

and matches the UV current as expected.

## 5.3 $\mathrm{SU}(3) \rightarrow [\mathrm{SU}(2) \times \mathrm{U}(1)]/\mathbb{Z}_2$

Next, we turn to a different $\mathrm{SU}(3)$ breaking pattern, which can arise from an adjoint Higgs vev that is gauge-equivalent to the special form

$$\begin{pmatrix} v & 0 & 0 \\ 0 & v & 0 \\ 0 & 0 & -2v \end{pmatrix}. \tag{80}$$

This preserves the gauge group $H = [\mathrm{SU}(2) \times \mathrm{U}(1)]/\mathbb{Z}_2$. We have $\pi_1(H) \cong \mathbb{Z}$, generated by a path in the covering space $\mathrm{SU}(2) \times \mathrm{U}(1)$ which goes from the origin to a center element of $\mathrm{SU}(2)$ and a square root of unity in $\mathrm{U}(1)$. For the chosen Higgs vev, we can take the unbroken $\mathrm{SU}(2)$ to be generated by the matrices $\tau^i$ in (64) and the $\mathrm{U}(1)$ to be generated by the Gell-Mann matrix $\lambda^8$.

We can realize the minimally charged monopole as an 't Hooft-Polyakov monopole embedded in the *lower* $2 \times 2$ block, which not only carries $\mathrm{U}(1)$ magnetic charge, but also magnetic charge under $\mathrm{U}(1) \subset \mathrm{SU}(2)$. Indeed, we worked out precisely this case with $\mathrm{M}'$ in the previous example! This monopole carries charge $-1/2$ under $\mathrm{U}(1) \subset \mathrm{SU}(2)$ (the gauge field $A_3$) and $\sqrt{3}/2$ under $\mathrm{U}(1)$ (the gauge field $A_8$). We can see that this monopole is allowed by the Dirac quantization condition as follows. If we focused on only the $\mathrm{SU}(2)$ part, then with a $2\pi$ gauge transformation we would have

$$\exp\left[2\pi i \times -\frac{1}{2}\tau^3\right] = \exp[-i\pi\tau^3] = -\mathbb{1}, \tag{81}$$

and within the U(1) part we would have

$$\exp\left[2\pi i \times \frac{\sqrt{3}}{2}\lambda^8\right] = \exp[i\pi\sqrt{3}\lambda^8] = -\mathbb{1}, \tag{82}$$

but their product is the identity matrix $\mathbb{1}$. This is exactly the minimal monopole we expect, taking into account the $\mathbb{Z}_2$ quotient.

Thus, we have essentially already worked everything out! In the case of SU(3) $\to$ U(1) $\times$ U(1), we had three potential Chern-Weil currents, $F_3 \wedge F_3$, $F_8 \wedge F_8$, and $F_3 \wedge F_8$. The last was eliminated by the monopole M, and only the linear combination of the first two matching the UV Chern-Weil current was preserved by the monopole M'. In the case of SU(3) $\to$ [SU(2) $\times$ U(1)]/$\mathbb{Z}_2$, the monopole M is absent, but we also only have two Chern-Weil currents in the IR theory: $\mathrm{tr}_{\mathrm{SU}(2)}(F \wedge F)$ and $F_8 \wedge F_8$. There is no analogue of the combination that was eliminated by M, and there is no monopole M. The argument that M' preserves only the correct linear combination of $\mathrm{tr}_{\mathrm{SU}(2)}(F \wedge F)$ and $F_8 \wedge F_8$ to match the UV current is exactly parallel to the previous example.

Still, there is slightly more to say here, because one might wonder: in this example, the IR theory contains additional gauge fields, the $A^1$ and $A^2$ gauge bosons of SU(2). Could the minimal monopole of this theory admit additional dyon modes that are charged under these generators? This question was investigated in [66–71], with the result that it does not—the only dyon mode of the minimal monopole of [SU(2) $\times$ U(1)]/$\mathbb{Z}_2$ is precisely the one we labeled $\sigma'$ in the discussion above. We will assume that this holds more generally, and dyon modes are never charged under generators outside of the Cartan subalgebra.

## 5.4  SU(5) $\to$ [SU(3) $\times$ SU(2) $\times$ U(1)]/$\mathbb{Z}_6$

Next, we turn to a very similar example, but one with great real-world interest: the breaking of an SU(5) GUT group to the Standard Model gauge group (with a $\mathbb{Z}_6$ quotient). Here $\pi_1(H) \cong \mathbb{Z}$ is generated by a path that, in the cover SU(3) $\times$ SU(2) $\times$ U(1), goes from the origin to a center element of SU(3) and SU(2) and a 6th root of unity in U(1).

For concreteness, consider the adjoint Higgs vev $\langle\Phi\rangle = \mathrm{diag}(2v, 2v, 2v, -3v, -3v)$. Then we can take the SU(3) generators to be the Gell-Mann matrices $\lambda^i$ embedded in the upper left $3 \times 3$ block and the SU(2) generators to be the Pauli matrices $\tau^i$ embedded in the lower right $2 \times 2$ block. The generator of hypercharge is

$$\tau_Y \equiv \frac{1}{\sqrt{15}}\begin{pmatrix} 2 & 0 & 0 & 0 & 0 \\ 0 & 2 & 0 & 0 & 0 \\ 0 & 0 & 2 & 0 & 0 \\ 0 & 0 & 0 & -3 & 0 \\ 0 & 0 & 0 & 0 & -3 \end{pmatrix}. \tag{83}$$

This is normalized so that $\mathrm{tr}(\tau_Y\tau_Y) = 2$, and we also have $\mathrm{tr}(\tau_Y\lambda^i) = 0$, $\mathrm{tr}(\tau_Y\tau^i) = 0$. Note that

$$\exp[2\pi i\sqrt{15}\tau_Y] = \mathbb{1}. \tag{84}$$

The $\mathbb{Z}_6$ quotient arises because

$$\exp\left[\frac{2\pi}{6}i\sqrt{15}\tau_Y\right] = \mathrm{diag}(e^{2\pi i/3}, e^{2\pi i/3}, e^{2\pi i/3}, e^{\pi i}, e^{\pi i}) = \exp\left[\frac{2\pi}{3}i\sqrt{3}\lambda^8\right]\exp\left[\frac{2\pi}{2}i\tau^3\right]. \tag{85}$$

This suggests that, under the generators $(\lambda^8, \tau^3, \tau_Y)$, our minimally charged monopole should carry magnetic charge $(-\frac{1}{3}\sqrt{3}, -\frac{1}{2}, \frac{1}{6}\sqrt{15}) = (-\frac{1}{\sqrt{3}}, -\frac{1}{2}, \frac{1}{2}\sqrt{\frac{5}{3}})$, in order to wind around the generator of $\pi_1(H)$.

We can construct an 't Hooft-Polyakov monopole by embedding the SU(2) generators of the 't Hooft-Polyakov solution in the third and fourth rows and columns, so that the monopole charge corresponds to

$$\sigma^3 = \begin{pmatrix} 0 & 0 & 0 & 0 & 0 \\ 0 & 0 & 0 & 0 & 0 \\ 0 & 0 & 1 & 0 & 0 \\ 0 & 0 & 0 & -1 & 0 \\ 0 & 0 & 0 & 0 & 0 \end{pmatrix}. \tag{86}$$

In terms of the generators $\lambda^8$, $\tau^3$, and $\tau_Y$, we have

$$\sigma^3 = -\frac{1}{\sqrt{3}}\lambda^8 - \frac{1}{2}\tau^3 + \frac{1}{2}\sqrt{\frac{5}{3}}\tau_Y. \tag{87}$$

This is precisely the linear combination we had inferred from the homotopy argument, so the 't Hooft-Polyakov monopole is a minimal-charge monopole for $H$. For clarity of notation, let us use $G_i$ for the SU(3) gauge fields, $W_i$ for the SU(2) gauge fields, and $A_Y$ for the U(1) gauge field. The fluxes sourced by the monopole are,

$$\int_\Sigma \frac{1}{4\pi}\, dG_8 = -\frac{1}{\sqrt{3}}, \qquad \int_\Sigma \frac{1}{4\pi}\, dW_3 = -\frac{1}{2}, \qquad \int_\Sigma \frac{1}{4\pi}\, dA_Y = \frac{1}{2}\sqrt{\frac{5}{3}}. \tag{88}$$

The dyon mode $\sigma$ corresponds to the charge of the broken generators of the SU(2) embedded in the third and fourth columns, which corresponds to the same linear combination, i.e., the covariant derivative is

$$D\sigma = d\sigma - \left(-\frac{1}{\sqrt{3}}G_8 - \frac{1}{2}W_3 + \frac{1}{2}\sqrt{\frac{5}{3}}A_Y\right), \qquad d(D\sigma) = \frac{1}{\sqrt{3}}\,dG_8 + \frac{1}{2}\,dW_3 - \frac{1}{2}\sqrt{\frac{5}{3}}\,dA_Y. \tag{89}$$

We can improve currents using integer multiples of the term $\frac{1}{2\pi}D\sigma \wedge \delta_M$. It is then clear that this can only be used to improve a Chern-Weil current that involves all three of the gauge groups. Indeed, we have

$$\begin{aligned} J &= \frac{1}{8\pi^2}\mathrm{tr}_{SU(5)}(F \wedge F) \\ &= \frac{1}{8\pi^2}\mathrm{tr}_{SU(3)}(dG \wedge dG) + \frac{1}{8\pi^2}\mathrm{tr}_{SU(2)}(dW \wedge dW) + \frac{1}{16\pi^2}dA_Y \wedge dA_Y + \cdots, \end{aligned} \tag{90}$$

and isolating the parts of this involving $G_8$ and $W_3$ and taking a derivative using (88),

$$\begin{aligned} &d\left[\frac{1}{16\pi^2}\,dG_8 \wedge dG_8 + \frac{1}{16\pi^2}\,dW_3 \wedge dW_3 + \frac{1}{16\pi^2}\,dA_Y \wedge dA_Y\right] \\ &= \frac{1}{2\pi}\left(-\frac{1}{\sqrt{3}}\,dG_8 - \frac{1}{2}\,dW_3 + \frac{1}{2}\sqrt{\frac{5}{3}}\,dA_Y\right) \wedge \delta_M. \end{aligned} \tag{91}$$

This shows that the unique linear combination of IR Chern-Weil currents that can be improved using the dyon mode is the one that descends from the SU(5) Chern-Weil current in the UV.

## 5.5 Other breaking patterns

### 5.5.1 Taking stock

We have seen several examples of GUT theories with simple and simply connected gauge group $G$ Higgsed down to $H$. Beyond the specific features of each example there is a common thread:

- The UV theory has an exact U(1) Chern-Weil symmetry that is enhanced to $U(1)^\alpha$ in the IR. The integer $\alpha$ depends on the details of the Higgsing pattern and is bounded above, $\alpha \le \text{rank}(G)$.

- GUT theories generically admit monopole solutions. Monopoles carry charges under generators of the Lie algebra. These charges are constrained to be mutually local with electric representations. The more representations, the more constraining this requirement is and the fewer monopoles are compatible with it. For instance, if $H$ is the universal cover, allowed monopoles are restricted to lie in the dual root lattice of $H^\vee$. On the other hand, if $H$ is the adjoint group, monopoles are less restricted and their charges lie in the algebra of the dual group $H^\vee$.

- If $H$ is not simply connected, a subset of these monopoles is charged under a magnetic symmetry $\pi_1(H)$.[19] The least energetic monopoles with a particular magnetic charge are therefore topologically stable.

- Monopoles may carry dyonic modes. In 4d these are compact scalar fields living on the monopole world-line. We have used the fact that dyons are only charged under the Cartan subalgebra of the Lie algebra [66–71].

- These monopole solutions are finite energy and, therefore, dynamical. The IR Chern-Weil current is generically fully broken when processes involving dynamical monopoles are taken into account. These processes are inherently non-perturbative and we restrict our analysis to processes involving topologically stable monopoles, which are under better control.

- Dyon modes can be used to recover a subgroup of the IR Chern-Weil symmetry. If enough stable monopoles are present the IR Chern-Weil symmetry is a diagonal combination of the Cartan subalgebra generators, matching the UV U(1) Chern-Weil symmetry. This follows from the dyon modes being charged only under the Cartan subalgebra.

All previous examples had enough stable monopoles to recover the UV Chern-Weil symmetry. It turns out that this feature is rather generic but not universal. It depends crucially on the local and global properties of $G$ and $H$. In the following section we explore these issues and lay out a procedure to find the IR Chern-Weil current that is conserved in the presence of dynamical monopoles with dyon modes.

### 5.5.2 Overview of a general procedure

Matters are simplified by considering a simply connected $G$, which we do in the following. In this case the IR Chern-Weil symmetry preserved by stable monopoles depends only on the local and global structure of $H$. Let us consider for definiteness $H$ to be a direct product,

$$H = H_1 \times H_2 \times \ldots \times H_n, \tag{92}$$

where each factor $H_i$ $(i = 1, \ldots, n)$ can't be further split into a direct product. Take each factor $H_i$ to have the generic form,

$$H_i = \frac{H_i^1 \times H_i^2 \times \ldots \times H_i^{m_i}}{\Gamma} . \tag{93}$$

---

[19]The topological classification of monopoles formed from higgsing is given by $\pi_2(G/H)$. If $G$ is simply connected (has vanishing fundamental group) an exact short sequence of homotopy groups implies $\pi_2(G/H) \cong \pi_1(H)$, matching the IR magnetic symmetry.

We take the $H_i^{\tilde{i}}$ to be simple and simply connected Lie groups or U(1)s, and $\Gamma$ is a discrete group, typically a cyclic group.[20] Unlike for other indices, we will not sum over repeated indices of the $i, \tilde{i}$ type, unless explicitly stated. In the following we explain the generic procedure to determine the Chern-Weil symmetry preserved by dynamical monopoles with dyon modes. We will make some comments about broad classes of models along the way. The analysis consists of three steps.

**1. The naive Chern-Weil current in the IR.**  Let us introduce field strengths $F_i^{\tilde{i}}$ for each $H_i^{\tilde{i}}$. It follows from the Bianchi identity that $d \operatorname{Tr}(F_i^{\tilde{i}} \wedge F_i^{\tilde{i}}) = 0$. For U(1) factors we can also have off-diagonal Chern-Weil currents. The most general IR (naively) conserved Chern-Weil current is a linear combination,

$$j = \frac{1}{8\pi^2} \sum_{i,\tilde{i}} \alpha_i^{\tilde{i}} \operatorname{Tr}(F_i^{\tilde{i}} \wedge F_i^{\tilde{i}}) + \frac{1}{8\pi^2} \sum_{i,\tilde{i} \neq \tilde{j} \in \mathrm{U}(1)\mathrm{s}} \beta_i^{\tilde{i},\tilde{j}} F_i^{\tilde{i}} \wedge F_i^{\tilde{j}} \,, \tag{94}$$

with coefficients $\alpha_i^{\tilde{i}}, \beta_i^{\tilde{i},\tilde{j}}$ whose quantization depends on the global form of $H$ and are generically simple rationals if $j$ is chosen to be integrally quantized. There is, a priori, no further restriction on the $\alpha_i^{\tilde{i}}, \beta_i^{\tilde{i},\tilde{j}}$. If we let $n_i$ denote the number of non-abelian $H_i^{\tilde{i}}$ factors for a given $i$, and $r_i$ the number of U(1) $H_i^{\tilde{i}}$ factors, the naive IR Chern-Weil symmetry is $\mathrm{U}(1)^{\sum_i (n_i + r_i(r_i+1)/2)}$.

**2. Dynamical monopoles break the current $j$.**  This symmetry is generically broken non-perturbatively by dynamical monopoles. Denote by $H^c$ the generators of the Cartan subalgebra of $h$ such that $c = 1, ..., r$ and $r = \operatorname{rank}(h)$. By a suitable choice of gauge, the Cartan subalgebra can be chosen to contain any given element in the Lie algebra. Consider a stable monopole $M$ with charge,

$$Q_M = 4\pi \sum_{c=1}^{r} (k_M)_c H^c \,. \tag{95}$$

The vector $(k_M)_c$ defines the weights of the monopole $M$ and denotes its charge under the generator of the Cartan subalgebra. If the monopole is charged under one or more of the generators of the Cartan subalgebra of $H_i^{\tilde{i}}$ the corresponding piece of the current $j$ is no longer conserved,

$$d \operatorname{Tr}(F_i^{\tilde{i}} \wedge F_i^{\tilde{i}}) \neq 0 \,. \tag{96}$$

In order to find the surviving conserved Chern-Weil current one must list all stable monopoles and their charges under the $H_i^{\tilde{i}}$. If some factors are uncharged under every monopole there will be an unbroken $j$. If each factor is charged under at least one stable monopole, no Chern-Weil current will be preserved. The details are model dependent, but we can make some broad comments.

- If $H$ contains at least one factor $H_i$ with vanishing fundamental group, no stable monopole will be charged under it and a conserved $J_{\mathrm{IR}}$ will remain.

- If $H$ contains no factor $H_i$ with vanishing fundamental group the opposite is true and no conserved $j$ remains. In particular, every $H_i^{\tilde{i}}$ will be charged under at least some stable monopole. This holds because the quotient $\Gamma$ must act on $H_i^{\tilde{i}}$, otherwise it would factor out of $H_i$. It follows that some element of $\pi_1(H_i)$ involves an element of $H_i^{\tilde{i}}$, implying the existence of a monopole topologically charged under $H_i^{\tilde{i}}$. The examples in sections 5.1 to 5.4 belong to this category.

---

[20]This choice of $H$ is general enough to cover all examples considered in this work. We expect our considerations to hold for more general $H$.

**3. Dyon modes and the improved $J_{IR}$.**  Dyon modes arising on the monopole worldlines can be used to define an improved IR current $J_{IR}$ that is conserved in the presence of dynamical monopoles even if $j$ is not. $J_{IR}$ will generically have more independent terms than $j$ that are conserved and will give rise to a bigger Chern-Weil symmetry.[21] As we have previously seen, monopoles in GUT theories generically carry dyon modes with gauge charges that match the monopole weights. Dyon modes arise from the spontaneous breaking of global gauge symmetries in the monopole vacuum. Global color is only well-defined for elements in the Lie algebra commuting with all magnetic charges [66–71]. Thus, dyon modes are not charged under Lie algebra generators outside the Cartan subalgebra. The dyon covariant derivative is then expressed as[22]

$$\frac{1}{2\pi}\left(\mathrm{d}\sigma_{\mathrm{M}}-\sum_c (\mathbf{k}_{\mathrm{M}})_c A_c\right)\wedge\delta_{\mathrm{M}}\,. \tag{97}$$

In the presence of the monopole $M$ we can consider improving $j$ with terms proportional to eq. (97) as long as the quantization of $j$ is preserved. If we choose $j$ to be integrally quantized we are free to add integer multiples of eq. (97). The objective now is to find the biggest improved current that is conserved in the presence of *any* stable monopole. This is what we call $J_{IR}$. As we saw in the previous sections this is a bit of an art and we leave a systematic exploration for the future.[23] We can make some general comments:

- Since dyon modes are only charged under the Cartan subalgebra, the improvement from $j$ to $J_{IR}$ can only "save" a current that is a linear combination of elements in the Cartan subalgebra. The coefficients in this linear combination are a priori independent.

- If a monopole is charged under several elements of the Cartan subalgebra, their corresponding coefficients in the improved current $J_{IR}$ will not be independent.

- If there are enough monopoles to break all of $j$, it may happen that there are also enough to constrain the coefficients in $J_{IR}$ such that a single linear combination of all Cartan algebra elements survives. In this case the IR Chern-Weil symmetry preserved by dynamical monopoles will be U(1). We will say that models with this property have *abundant monopoles*. A necessary but not sufficient condition for this is as we discussed in step 2: $H$ contains no factor $H_i$ with $\pi_1(H_i)=0$. For an argument that this is not sufficient, see Sec. 5.6.4.

- Furthermore, in a model with abundant monopoles, the U(1) Chern-Weil symmetry in the IR will match the one in the UV. Whenever the IR EFT reproduces the UV Chern-Weil symmetries, we dub this phenomenon *Chern-Weil symmetry matching*. All the examples we saw so far belong to this group.

### 5.5.3 Some families with Chern-Weil symmetry matching

We have seen that, generically, the Chern-Weil symmetries of the UV and the IR theories may differ, depending on the ranks of $G,H$ and the amount of stable monopoles. If there are abundant monopoles, the Chern-Weil symmetries match. In this section we discuss some families of

---

[21]Consider for instance a model with naive $j$ giving rise to a Chern-Weil symmetry $U(1)^a$. Dynamical monopoles will break some of these U(1)'s and the non-perturbatively conserved current will be $U(1)^{a'}$, with $a\geq a'$. The improved current $J_{IR}$ is such that, in the presence of dynamical monopoles, a bigger symmetry survives $U(1)^b$ with $a\geq b\geq a'$.

[22]This expression can be explicitly computed in a given model. We give two examples in appendix A.

[23]A problem that arises when trying to carry out a systematic treatment is that eq. (95) holds for a single monopole. It is generically not possible to find a gauge such that all monopole charges lie in the Cartan subalgebra.

models that satisfy this property. We remain agnostic about the UV but assume that the rank is not reduced.[24]

The first such family is $H = [SU(N-1) \times U(1)]/\mathbb{Z}_{N-1}$. A minimal stable magnetic monopole has charge $\frac{1}{(N-1)}$ under U(1) and a generator of the Cartan subalgebra of SU($N-1$). A suitable Weyl reflection maps this monopole solution to an inequivalent stable monopole charged under any other element of the Lie algebra of SU($N-1$).[25] These monopoles fully break $j$ and a linear combination matching the UV Chern-Weil symmetry can be saved using the dyon modes. The models in sections 5.1 and 5.3 belong to this group.

A similar family is $[SU(M) \times SU(N-M) \times U(1)]/\mathbb{Z}_{\text{LCM}(M,N-M)}$. In this case a minimal stable monopole will carry charge $\frac{1}{\text{LCM}(M,N-M)}$ under U(1) and $\frac{1}{M}$, $\frac{1}{N-M}$ under some root of SU($M$) and SU($N-M$), respectively. A Weyl rotation inside SU($M$) and SU($N-M$) maps this solution to other minimal stable monopoles. These are again sufficient to conclude that the Chern-Weil symmetry is matched. The model in section 5.4 belongs to this group.

## 5.6 More examples

We have extensively discussed models exhibiting Chern-Weil symmetry matching, but this matching is not universal. In this section, we first consider examples with the same group $G$ as in previous cases, but where rank($G$) > rank($H$). Then, we explore cases with other simple and simply connected $G$. In some of these instances, we find no Chern-Weil symmetry matching due to the absence of abundant monopoles. Further examples with non-simple groups are discussed in Appendix B.

### 5.6.1 SU(5) → SU(3) × SU(2)

Let us first explore an example where the rank is reduced with SU(5) → SU(3) × SU(2). Consider an SU(5) gauge theory with two Higgs fields, one in the adjoint Higgs and another in the **10** representation. The adjoint Higgs field induces the symmetry breaking SU(5) → [SU(3) × SU(2) × U(1)]/$\mathbb{Z}_6$, the Standard Model gauge group. We then focus on the condensation of the SU(3) and SU(2) singlet component within the **10** representation (a "right-handed selectron"), resulting in further symmetry breaking to [SU(3) × SU(2) × $\mathbb{Z}_6$]/$\mathbb{Z}_6 \cong$ SU(3) × SU(2). Notably, this scenario does not support any truly stable monopoles or strings due to the triviality of the homotopy groups $\pi_2(G/H) = 0$ and $\pi_1(G/H) = 0$, where $G = $ SU(5) and $H = $ SU(3) × SU(2). If the **10** VEV is much smaller than the adjoint VEV, each GUT monopole is attached to a string, and each string can be broken via the creation of a monopole-antimonopole pair. These configurations are quite similar to the ones obtained from SU(2) → U(1) → 1 involving both the adjoint and fundamental representation Higgs fields, as reviewed in [73].

In this scenario where stable monopoles are absent, we find that two IR CW currents remain conserved, corresponding to the instanton number of the SU(2) and SU(3) gauge groups. On the other hand, there is only one UV CW current associated with the SU(5) symmetry. This example illustrates that the conservation of IR CW currents does not necessarily align with those in the UV regime when the rank is reduced. One of the two IR currents is emergent, and is explicitly broken by UV effects, but this breaking is not visible in terms of the IR effective theory that we are considering.

---

[24]We also continue to assume that $G$ has vanishing fundamental group.

[25]Equivalently, these different monopoles arise from different embeddings of SU(2) in $G$ [72].

### 5.6.2 SU(3) → SO(3)

Next, let us study the example SU(3) → SO(3) in which the rank is reduced. The spontaneous symmetry breaking of SU(3) → SO(3) is obtained by the condensation of the 6-representation scalar field $S$ which is expressed with a $3 \times 3$ symmetric matrix,

$$\langle S \rangle = v \begin{pmatrix} 1 & 0 & 0 \\ 0 & 1 & 0 \\ 0 & 0 & 1 \end{pmatrix}. \tag{98}$$

The generators of the SO(3) are

$$\lambda_2 = \frac{1}{2} \begin{pmatrix} 0 & -i & 0 \\ i & 0 & 0 \\ 0 & 0 & 0 \end{pmatrix}, \qquad -\lambda_5 = \frac{1}{2} \begin{pmatrix} 0 & 0 & -i \\ 0 & 0 & 0 \\ i & 0 & 0 \end{pmatrix}, \qquad \lambda_7 = \frac{1}{2} \begin{pmatrix} 0 & 0 & 0 \\ 0 & 0 & -i \\ 0 & i & 0 \end{pmatrix}, \tag{99}$$

where the first generator $\lambda_2$ is chosen to be the generator of the Cartan subalgebra. In this breaking pattern, we have a $\mathbb{Z}_2$ monopole because of

$$\pi_2[\mathrm{SU}(3)/\mathrm{SO}(3)] = \pi_1[\mathrm{SO}(3)] = \pi_1[\mathrm{SU}(2)/\mathbb{Z}_2] = \mathbb{Z}_2\,.$$

As in (62), the UV Chern-Weil current is

$$J = \frac{1}{8\pi^2} \mathrm{tr}(F \wedge F) = \frac{1}{16\pi^2}[F_2 \wedge F_2 + \cdots]\,, \tag{100}$$

where $\cdots$ denotes the other $A_i$ including both massive and massless component.

For one monopole, we have

$$\frac{1}{2} \frac{1}{2\pi} \int_\Sigma F_2 = 1\,, \tag{101}$$

where the factor $1/2$ comes from the normalization of $\mathrm{tr}[T^i T^i] = 1/2$. The covariant derivative of the dyon mode $\sigma$ is expressed as

$$D\sigma = \mathrm{d}\sigma - A_2\,, \tag{102}$$

as discussed in more detail in Appendix A.2. The IR current is unique owing to the SO(3) non-abelian group and recovered by the dyon mode,

$$J_{\mathrm{IR}} = \frac{1}{16\pi^2}(F_2 \wedge F_2 + F_5 \wedge F_5 + F_7 \wedge F_7) + \frac{1}{2\pi} D\sigma \wedge \delta_{\mathrm{M}}\,, \tag{103}$$

which matches with the UV current.

### 5.6.3 Spin(10) → [SU(4) × SU(2) × SU(2)]/$\mathbb{Z}_2$

Now, let us discuss an example in SO(10) GUT. In the breaking of

$$\mathrm{Spin}(10) \rightarrow [\mathrm{SU}(4) \times \mathrm{SU}(2) \times \mathrm{SU}(2)]/\mathbb{Z}_2\,,$$

we have a $\mathbb{Z}_2$ monopole because of $\pi_1([\mathrm{SU}(4) \times \mathrm{SU}(2) \times \mathrm{SU}(2)]/\mathbb{Z}_2) = \mathbb{Z}_2$. The magnetic charge of the stable monopole is given with the generators of SU(4) and either SU(2) because of the $\mathbb{Z}_2$ quotient, which leads to the Chern-Weil symmetry matching.

### 5.6.4  USp(4) → SU(2) × SU(2)

For $G = \text{USp}(2n)$, we find examples in which $G$ and $H$ have the same rank but the number of the conserved Chern-Weil currents are different between UV and IR. For instance, we have $\text{USp}(4) \cong \text{Spin}(5) \to \text{Spin}(4) \cong \text{SU}(2) \times \text{SU}(2)$ by the condensation of a Higgs in the **5** representation of USp(4). Note that there is no quotient because the fundamental representation of USp(4) is decomposed as $\mathbf{4} = (\mathbf{2},\mathbf{1}) \oplus (\mathbf{1},\mathbf{2})$ preventing a $\mathbb{Z}_2$ quotient. Besides, we do not obtain any monopoles because of $\pi_2(G/H) = \pi_1(H) = 0$. Therefore, two IR Chern-Weil currents are conserved, while there is only one UV Chern-Weil current. As in Sec. 5.6.1, this means that one of the two IR currents is emergent and broken by UV effects, but we do not have an interpretation of this breaking in terms of the EFT of defects within the IR theory.

We can further break $\text{SU}(2) \times \text{SU}(2) \to \text{U}(1) \times \text{U}(1)$ with a generic VEV of the adjoint Higgs of USp(4). Consider the case where the VEV of the adjoint Higgs is much smaller than the VEV of the **5** Higgs, so that we regard there are two independent SU(2)'s in the IR theory and each SU(2) breaks into U(1).[26] Now, the story is quite similar to the previous discussion of $\text{SU}(2) \to \text{U}(1)$. A stable monopole charged under each U(1) leads to two improved IR Chern-Weil currents.

On the other hand, we could have $\text{USp}(4) \to \text{U}(1) \times \text{U}(1)$ in a different way. With a condensation of the adjoint Higgs, we can have $\text{USp}(4) \to [\text{SU}(2) \times \text{U}(1)]/\mathbb{Z}_2$ [74]. Due to the overall quotient, a stable monopole is charged under both generators of SU(2) and U(1), which indicates there exits only one improved IR Chern-Weil current corresponding to the UV one. We also consider a further symmetry breaking to $\text{U}(1) \times \text{U}(1)$ with a small VEV of the **5** Higgs.[27] Due to the smallness of the VEV, the core configuration of the monopole will not be affected by the second breaking. This indicates the monopole is still stable, and there exits only one improved IR Chern-Weil current.

Here is an apparent contradiction regarding the number of improved IR Chern-Weil currents between the two symmetry breaking pathways, despite both involving condensations of the adjoint and **5** Higgs fields. A potential resolution to this contradiction is that the stability of certain monopoles changes somewhere in the parameter space of the theories as the hierarchy between the VEVs of the adjoint and **5** Higgs fields reverses. More concretely, some monopoles that are stable in the second scenario become unstable in the intermediate region toward the first scenario, resulting in two improved IR Chern-Weil currents. A detailed investigation into the stability of these monopoles is left for future research.

## 6  Discussion and conclusions

The Chern-Weil global symmetries of gauge theory, such as the instanton number $(d-5)$-form symmetry, are robust within low-energy effective field theory, because they follow from the Bianchi identity of the gauge group. Nonetheless, there should be mechanisms for eliminating such global symmetries, at least in the context of quantum gravity. The most well-understood route to eliminating Chern-Weil symmetries is by gauging them. This can be accomplished through a Chern-Simons coupling to a dynamical gauge field, or by other couplings that render the Chern-Weil symmetry current *exact* instead of merely conserved. An example of the latter case, in $d = 4$, is coupling to fermions with a chiral $\text{U}(1)_\text{C}$ symmetry that has an ABJ anomaly with the gauge fields, leading to an equation of the form $dJ_\text{C} = \frac{1}{8\pi^2}\text{tr}(F \wedge F)$ [9]. As emphasized in [30], in the 2d case gauging the current $\frac{1}{2\pi}F$ by coupling to a dynamical axion and gauging by coupling to anomalous chiral fermions are related through the bosonization dictionary.

---

[26]The adjoint Higgs of USp(4) contains the adjoint Higgses for both SU(2)'s.

[27]This also affects the VEV of the adjoint Higgs, leading to a generic adjoint VEV.

The explicit breaking of Chern-Weil symmetries is much less well understood than their gauging. Different mechanisms of explicitly breaking such symmetries were discussed in [9], as well as examples in string theory of Chern-Weil symmetries that were not gauged by Chern-Simons terms but for which the breaking mechanism was not well understood. In this paper, we have clarified the mechanism of explicit breaking of Chern-Weil symmetries by magnetic monopoles, showing that it is more widespread than previously argued in [9], and in particular that it can apply to non-abelian gauge theories. Quite generally, when a gauge group $G$ has nontrivial fundamental group $\pi_1(G)$, magnetic monopoles can exist (and *should* exist in quantum gravity, by the completeness hypothesis [50,52]) and violate the Bianchi identity for the gauge group. This breaking of the Bianchi identity causes other Chern-Weil currents to be broken as well. We have discussed this explicitly for PSU($n$) gauge theory in Sec. 4.

In many cases, this unifies two apparently different mechanisms for breaking Chern-Weil symmetries that were discussed in [9]. That reference discussed explicit breaking of instanton number symmetry by magnetic monopoles for U(1) gauge theory, and explicit breaking of multiple instanton number symmetries in the IR to a single UV symmetry in the case of higgsing (e.g., for SU(5) to $G_{\mathrm{SM}}$), by shrinking instantons below the higgsing scale and then rotating them within the larger gauge group. In Sec. 5, we have seen that often these are the same mechanism: under a higgsing pattern $G \to H$, there are magnetic monopoles for $H$ associated with a nontrivial $\pi_2(G/H)$, which explicitly break the Chern-Weil symmetries of $H$. However, these magnetic monopoles can admit dyon degrees of freedom, and we have shown that in many cases these dyon degrees of freedom allow for an improved Chern-Weil symmetry current for $H$, including localized contributions on monopole worldlines, that is conserved and matches the UV Chern-Weil symmetry of $G$.

It would be appealing if we could argue that this mechanism is universal, but it is not. For example, we have seen examples in Sec. 5.6.1 and Sec. 5.6.4 of a higgsing pattern $G \to H$ where $G$ has a single instanton number symmetry, $H$ has two different instanton number symmetries, and $\pi_1(H)$ is trivial, so there are no monopoles to break the larger IR symmetry to the smaller UV one. This counterexample shows that it is not always possible, within the low-energy gauge theory, to identify a conserved charge associated with a possible dynamical object whose existence would explicitly break the emergent infrared Chern-Weil symmetry. (There may be dynamical objects, such as confined monopoles, that do so; however, we cannot infer their existence from the IR gauge group alone, because they do not have an associated conserved charge.)

The situation may be better in quantum gravity. First, let us briefly recap other examples of explicitly broken Chern-Weil symmetries in string theory from [9], in light of our current understanding. One familiar example is the $\mathrm{AdS}_5 \times S^5$ compactification of Type IIB string theory. Viewed as a 5d AdS quantum gravity theory, this theory has an apparent SO(6) gauge symmetry arising from isometries of the 5-sphere, but there is no corresponding $C \wedge \mathrm{tr}(F \wedge F)$ Chern-Simons coupling in the effective action. Thus, $\mathrm{tr}(F \wedge F)$ for SO(6) appears to be a current generating a good 0-form global symmetry of the theory, which should somehow be explicitly broken by UV dynamics. To see whether this symmetry could be explicitly broken by magnetic monopoles, we must identify the correct global form of the gauge group. It is well known that in fact, this is not SO(6) but its double cover Spin(6) $\cong$ SU(4), which has trivial fundamental group, and hence no magnetic monopoles. However, the situation is somewhat more subtle, as is clear from the dual field theory where SU(4) is an $R$-symmetry that rotates the four gauginos of $\mathcal{N} = 4$ SYM into each other. All fields that transform in representations of SU(4) that are not representations of SO(6), in fact, are fermions. Thus, the full symmetry group is a quotient of the product of the spacetime symmetry of $\mathrm{AdS}_5$ and the Spin(6) gauge symmetry, i.e., $[\mathrm{Spin}(4,2) \times \mathrm{Spin}(6)]/\mathbb{Z}_2$. This means that there are no magnetic monopoles for the gauge group itself, but there should be more subtle $\mathbb{Z}_2$ spacetimes that reflect this quotient

structure. We expect that these can be understood as the dynamical objects that explicitly break the Chern-Weil symmetry, and leave a detailed investigation for future work.

Another example from [9] is the heterotic string theory with gauge group $[E_8 \times E_8] \rtimes \mathbb{Z}_2$, where the $\mathbb{Z}_2$ acts by exchanging the two copies of $E_8$. This theory would have two instanton number symmetries, one for each $E_8$, but in fact the sum of the two currents is gauged (by the field $B_6$) and the difference is explicitly broken. The difference of currents is, in fact, not gauge invariant under the $\mathbb{Z}_2$. The theory has a dynamical object that makes this manifest, namely the twist vortex (or 7-brane) whose holonomy exchanges the $E_8$ factors [52] (recently discussed in more detail in [75]).[28]

Given these considerations, it appears to be an open possibility that, whenever a Chern-Weil symmetry is explicitly broken in quantum gravity, there is some charge in the theory that is associated with a heavy object that obstructs the gauging of the Chern-Weil symmetry. In general, "charge" in this context should be understood in terms of cobordism classes [76]. It would be interesting to revisit the examples of [9] in more detail, and to consider new examples, from this viewpoint. Other examples involve the Chern-Weil symmetries associated with the tangent bundle of spacetime itself, i.e., cases where the field strength $F$ is replaced by the Riemann curvature two-form. These are relatively less well-studied and may shed light on the general story.

There is another common class of defect that we have not discussed so far that may play an important role in this story, namely, a codimension-1 end-of-the-world brane. These objects are familiar from the case of heterotic M-theory [77,78], and are expected to exist in quantum gravity much more generally [76]. In the $d = 4$ case, the existence of end-of-the-world branes poses an obvious possible obstruction to coupling a theory to an axion: under the operation $\theta \mapsto \theta + 2\pi$, we have

$$\exp\left[i\frac{\theta}{8\pi^2}\int_M \text{tr}(F \wedge F)\right] \mapsto \exp\left[i\frac{\theta}{8\pi^2}\int_M \text{tr}(F \wedge F)\right]\exp\left[i\frac{1}{4\pi}\int_{\partial M} K_{\text{CS}}\right], \quad (104)$$

where $K_{\text{CS}}$ is the Chern-Simons three-form.[29] Thus, if the boundary conditions leave the gauge field free to vary (and in particular, to admit nonzero values of $\int_{\partial M} K_{\text{CS}}$) on the end-of-the-world brane on $\partial M$, the theory is not invariant under background gauge transformations of the axion and the instanton number symmetry is explicitly broken. This could be remedied by dynamics on the end-of-the-world brane, much as improved Chern-Weil currents can be constructed for monopoles. For example, if the path integral of the theory sums over all possible Chern-Simons levels on the end-of-the-world brane, then the full path integral will remain invariant under $\theta \mapsto \theta + 2\pi$, with a monodromy rearranging the contributions of different Chern-Simons levels. The theory on the brane could, for instance, admit dynamical domain walls separating regions with different Chern-Simons level, with chiral fermions (edge modes) living on these walls due to anomaly inflow. We leave a more detailed look at whether end-of-the-world branes explicitly violate Chern-Weil symmetries in known theories of quantum gravity for future work.

Symmetries, and their absence in quantum gravity, can provide powerful tools for understanding physics. In particular, if quantum gravity indeed forbids $(-1)$-form instanton number symmetries (an idea that is closely linked to the absence of free parameters in quantum gravity), then we could argue that the instanton number symmetries of the Standard Model must be

---

[28]This is not fully satisfying, because one can still ask why the theory does not have a $\mathbb{Z}_2$-odd gauge field $B_6'$ coupling to the difference of instanton numbers. MR thanks Jake McNamara for emphasizing this point.

[29]One can discuss a magnetic monopole in similar terms. If we view the monopole not as pointlike but as a small ball, then by inserting it we are cutting a hole in spacetime with boundary of topology $S^2 \times \mathbb{R}$. Then integrating the Chern-Simons form $A \wedge F$ using the flux of $F$ over $S^2$ leaves a factor of $A$ integrated over the worldline $\mathbb{R}$. This is one way to think about the Witten effect.

gauged or explicitly broken. Their gauging would imply the existence of an axion, or a massless chiral fermion, which are familiar solutions to the Strong CP problem. It has been less clear what it would mean for Chern-Weil symmetries to be explicitly broken. We have taken steps toward clarifying the possible mechanisms of explicit breaking of such symmetries, but a full classification of such mechanisms and investigation of their potential real-world consequences remains an open challenge.

## Acknowledgments

We especially thank Daniel Aloni for collaboration in the early stages of this work, and for numerous useful discussions. We wish to thank Jeremías Aguilera, Andrea Antinucci, Daniel Brennan, Clay Córdova, Iñaki García-Etxebarria, Thomas Grimm, Jake McNamara, Marco Serone and Stathis Vitouladitis for useful discussions. EGV also thanks the Simons Center for Geometry and Physics and its Summer Physics Workshop, Universite Libre de Bruxelles and Oviedo University for kind hospitality at different stages of this work.

**Funding information** The work of EGV has been supported by Margatita Salas award CA1/RSUE/2021-00738, MIUR PRIN Grant 2020KR4KN2, INFN Iniziativa Specifica ST&FI and by COST action CA22113. MR is supported in part by the DOE Grant DE-SC0013607. MS is supported by JSPS KAKENHI Grant Numbers JP22J00537.

## A Dyon action

We derive the action for the dyon degree of freedom, crucial for restoring the IR CW symmetries that is explicitly broken by monopoles. We focus on the term obtained from the $\theta$-term,

$$S \supset \int d^4x \frac{\theta}{64\pi^2} G^a_{\mu\nu} G^a_{\rho\sigma} \epsilon^{\mu\nu\rho\sigma} , \tag{A.1}$$

where $G^a_{\mu\nu}$ denotes the field strength for a simple and simply connected gauge group. Specifically, we derive the dyon action in the form of

$$S \supset \int \frac{\theta}{2\pi} \left( d\sigma - \frac{2}{4\pi} \text{Tr}[AQ_M] \right) \wedge \delta_M , \tag{A.2}$$

which also gives the covariant derivative for dyons. The derivation proceeds through several steps. First, we study an 't Hooft-Polyakov monopole emerging from the symmetry breaking of SU(2) → U(1) with an adjoint Higgs VEV. Second, we explore the 't Hooft-Polyakov monopoles arising from breaking of SU(3) → SO(3) with the condensation of the Higgs in the six representation. Refer to the main text for detailed notation.

### A.1 SU(2) → U(1)

We first focus on the spontaneous symmetry breaking of SU(2) → U(1) induced by the non-zero vacuum expectation value of the adjoint scalar field $\phi = \phi^a T^a$. This was the setting in which the dyon mode was first studied [37,38,79]. To obtain the zero mode (dyon mode), we consider fluctuations around a monopole background,

$$A^a_\mu = A^{(0)a}_\mu + \delta A^a_\mu , \qquad \phi^a = \phi^{(0)a} + \delta\phi^a , \tag{A.3}$$

where the superscript (0) denotes the monopole background configuration. The dyon mode $\sigma$ is given as a collective coordinate related to the global U(1) rotation.[30] This is expressed as

$$
\begin{aligned}
U &= \exp(i\sigma\phi^{(0)}/v), \\
A_\mu^{(0)} &\to U A_\mu^{(0)} U^\dagger + i U(\partial_\mu U^\dagger), \\
\phi^{(0)} &\to U\phi^{(0)} U^\dagger,
\end{aligned}
\tag{A.4}
$$

where $v$ is a dimensionful parameter ensuring the phase is dimensionless. The choice of $v$ does not affect the physics since an apparent ambiguity in $v$ is removed by redefining $\sigma$. We take $v$ so that a surface integral of (A.9) gives the monopole charge. Under the rotation, we identify

$$
\delta A_\mu^a = \frac{\sigma}{v}\partial_\mu\phi^{(0)a}, \quad \delta\phi = 0,
\tag{A.5}
$$

where higher order terms in $\sigma$ are ignored. Note that physical states are only affected by the global transformation at spatial infinity, and $\sigma$ and $\sigma + 2\pi$ are equivalent there. See, e.g., [72, 80] for more detailed pedagogical discussion about the dyon collective coordinate.

Let us derive $\int\frac{\theta}{2\pi}d\sigma$. Choosing the gauge $A_0^a = 0$, the $\theta$-term is expressed as

$$
\int dt \int d^3x \frac{\theta}{8\pi^2}\dot{A}_i^a B_i^a,
\tag{A.6}
$$

where $B_i^a = -\frac{1}{2}\epsilon_{ijk}G_{jk}^a$. The separation of time and spatial integrations is for later convenience. The dyon mode $\sigma$ emerges upon substituting $\dot{A}_i^a$ with $\dot{\sigma}\delta A_i^a/\delta\sigma$, resulting in

$$
\int dt \frac{\theta}{8\pi^2}\dot{\sigma}\int d^3x \frac{\mathcal{D}_i\phi^{(0)a}}{v}B_i^{(0)a},
\tag{A.7}
$$

where (A.5) is used. At this step, we have transformed the derivative into a covariant derivative.[31] Applying integration by parts and utilizing the condition $\mathcal{D}_i B_i^{(0)a} = 0$, we obtain

$$
\int dt \frac{\theta}{8\pi^2}\dot{\sigma}\int d^2S_i \frac{1}{v}B_i^{(0)a}\phi^{(0)a}.
\tag{A.8}
$$

The surface integral in this expression yields the monopole charge,

$$
\int d^2S_i \frac{1}{v}B_i^{(0)a}\phi^{(0)a} = 4\pi,
\tag{A.9}
$$

leading to

$$
\int dt \frac{\theta}{2\pi}\dot{\sigma}.
\tag{A.10}
$$

Therefore, we derive the action,

$$
\int \frac{\theta}{2\pi}d\sigma \wedge \delta_M,
\tag{A.11}
$$

which preserves Lorentz invariance.

Let us consider an additional contribution to the dyon action from the $\theta$-term as described in Eq. (A.2). Without employing the gauge fixing condition $A_0^a = 0$, the $\theta$-term can be expanded to include

$$
\int dt \frac{\theta}{32\pi^2}\int d^3x(-4\mathcal{D}_i A_0^a)B_i^a = \int dt\left(\frac{-\theta}{4\pi^2}\right)\int d^2S_i \, \mathrm{Tr}[A_0 B_i],
\tag{A.12}
$$

---

[30]$U$ is not close to unity at spatial infinity.

[31]The background gauge condition can be satisfied with this transformation (see, e.g., [72, 80]).

where we used the Bianchi identity $\mathcal{D}_i B_i = 0$. This derivation has thus far been conducted for general forms of $A_0$ and $B_i$, while we finally adopt the gauge choice $A_0^a = 0$ and set $B_i$ as the monopole background configuration. Treating $A_0^a$ as a constant leads to

$$\int dt \left(\frac{-\theta}{4\pi^2}\right) \text{Tr}[A_0 Q_{\text{M}}] \int d^2 S_i \frac{\hat{r}_i}{4\pi r^2} = \int dt \left(\frac{-\theta}{4\pi^2}\right) \text{Tr}[A_0 Q_{\text{M}}] \left(= -\theta \frac{1}{2\pi} A_0^a k^a\right), \quad \text{(A.13)}$$

where the surface integral is evaluated with $B_i = Q_{\text{M}} \frac{\hat{r}_i}{4\pi r^2}$, assuming a large radius $r$ of the sphere. Consequently, the Lorentz-invariant form of the dyon term is expressed as

$$\int \theta \frac{1}{2\pi} \left(d\sigma - \frac{2}{4\pi} \text{Tr}[A Q_{\text{M}}]\right), \quad \text{(A.14)}$$

noting that the factor of 2 preceding the trace originates from the normalization condition $\text{Tr}[T^a T^b] = 1/2\delta^{ab}$.

## A.2  SU(3) → SO(3)

In the breaking of SU(2) → U(1) with the adjoint Higgs, we have derived the action of the dyon mode in the BPS limit in which the potential does not contribute to the total energy and its only role is to give the boundary condition for the Higgs VEV. In this limit, the monopole solution is obtained from the Bogomol'nyi equation which minimizes the energy for a given magnetic charge.

In more generic cases, there is no apparent Bogomol'nyi equation because the Higgs field now is not in the adjoint representation. However, [81] describes a method to find the corresponding Bogomol'nyi equation similar to the case of SU(2) → U(1). More concretely, let us consider a monopole from a generic breaking pattern of $G \to H$. The Higgs field may not be in the adjoint representation of $G$, but we may find a subgroup $\tilde{G} \subset G$ such that the Higgs field includes the adjoint representation of $\tilde{G}$. Note that $\tilde{G}$ may not be the same as $H$. For example, in SU(3) → SO(3) (i.e., $G = $ SU(3), $H = $ SO(3)), we find $\tilde{G} = $ SU(2) under which the Higgs field of the **6** representation of SU(3) is decomposed to **3** ⊕ **2** ⊕ **1**. The monopole solution is constructed for the adjoint (**3**) and singlet (**1**) components of the Higgs field, while the field values of the other components are zero. The singlet components have constant field values, otherwise there is no cancellation for the covariant derivative (such a cancellation is needed for the finite energy configuration). In this manner, the minimization of the energy leads to the Bogomol'nyi equation for that adjoint part of the Higgs. To obtain such a Bogomol'nyi equation, any other components except for the adjoint and singlet ones should have zero field values. [81] has examined various examples and conjectured that this requirement is met when using the minimal representation Higgs field that achieves the breaking $G \to H$. Following this approach, the dyon action can be derived similarly to the case of SU(2) → U(1). Indeed, the global U(1) rotation is obtained by $U = \exp(i\sigma\phi_{\text{adj},\tilde{G}}/v)$ where $\phi_{\text{adj},\tilde{G}}$ denotes the adjoint Higgs under $\tilde{G}$, and the other parts are ignored due to beging singlets under $\tilde{G}$ or having zero background field values. Focusing on the $\theta$-term for the $\tilde{G}$ gauge group, the Bogomol'nyi equation leads to the dyon action in the form of (A.2).

Let us demonstrate the derivation of the dyon action using the example of SU(3) → SO(3). We start by reviewing the concrete procedure to obtain the monopole solution of SU(3) → SO(3) as detailed in [82, 83], and confirm that this is indeed a working example of the method described above. Consider the SU(3) gauge theory with the Higgs field in the **6** representation, expressed as a symmetric tensor $\Phi^{ab}$ ($\Phi$ denotes the $3 \times 3$ symmetric matrix). $\Phi$ is transforming under a SU(3) gauge transformation $U$ by

$$\Phi \to U\Phi U^T, \quad \text{(A.15)}$$

where the superscript $T$ denotes the transpose. The vacuum expectation value of the Higgs field is taken to be

$$\Phi_{\text{string}}^{ab} = v\,\delta^{ab}\,, \tag{A.16}$$

in the string gauge. The monopole is "associated with" the SU(2) subgroup, with generators defined as

$$T^1 = \frac{1}{2}\lambda^3\,, \qquad T^2 = \frac{1}{2}\lambda^1\,, \qquad T^3 = \frac{1}{2}\lambda^2\,, \tag{A.17}$$

where $\lambda^i$'s are the Gell-Mann matrices, and $T^3$ is one of the generators of SO(3). Note that this SU(2) corresponds to $\tilde{G}$ in the previous discussion. The gauge field configurations in the string gauge are

$$A_r^{\text{string}} = A_\theta^{\text{string}} = 0\,, \qquad A_\phi^{\text{string}} = \frac{1}{g\,r}T^3\tan\left(\frac{\theta}{2}\right)\,, \tag{A.18}$$

where $g$ is a gauge coupling constant of SU(3), and $r, \theta, \phi$ are used for a spherical coordinate system. To transition from the string gauge to the hedgehog gauge we perform the gauge transformation,

$$U(\theta, \phi) = \exp(-i\phi T^3)\exp(-i\theta T^2)\exp(i\phi T^3)\,. \tag{A.19}$$

In the hedgehog gauge, the Higgs field configuration is

$$\Phi = vU(\theta, \phi)U^T(\theta, \phi) = v\begin{pmatrix} \cos\theta + i\sin\theta\sin\phi & -i\sin\theta\cos\phi & 0 \\ -i\sin\theta\cos\phi & \cos\theta - i\sin\theta\sin\phi & 0 \\ 0 & 0 & 1 \end{pmatrix}\,. \tag{A.20}$$

The gauge fields become

$$A_\mu = A_\mu^a T^a = \frac{1}{gr}\epsilon_{\mu ab}T^a\hat{r}_b\,, \tag{A.21}$$

where $\epsilon_{0ij} = 0$, $\hat{r}_a$ is the unit vector, and $\sum_{a=1}^3\hat{r}_a\hat{r}_a = 1$. The asymptotic form of the Higgs and gauge fields must approach these configurations. The Ansatz for the solution is

$$\Phi = \frac{1}{gr}\begin{pmatrix} \psi_1(r)(z+iy) & \psi_1(r)(-ix) & 0 \\ \psi_1(r)(-ix) & \psi_1(r)(z-iy) & 0 \\ 0 & 0 & \psi_2(r), \end{pmatrix}\,, \tag{A.22}$$

$$A_\mu = -\frac{K(r)-1}{gr}\epsilon_{\mu ab}T^a\hat{r}_b\,, \tag{A.23}$$

where $\psi_i(r)$ are functions that depends on the radial coordinate $r$, while $K(r)$ is another radial function determining the gauge field configuration.

After substituting the Ansatz into the action, the variation with respect to $\psi_{1,2}$ and $K$ leads to the equations of motion (EOMs). From the EOMs, it turns out that $\psi_2(r)$ is a constant in the BPS limit, which indicates that $\psi_2(r)$ corresponds to the singlet under $\tilde{G}$.[32] The remaining non-trivial $2 \times 2$ matrix of $\Phi$ behaves as the adjoint representation of $\tilde{G} = $ SU(2). This can be quickly checked by noting $\Phi$ is made by a symmetric combination of $\mathbf{3} \otimes \mathbf{3} = \mathbf{6} \oplus \bar{\mathbf{3}}$, and $\mathbf{3}$ is decomposed to $\mathbf{2} \oplus \mathbf{1}$ for SU(2). This indicates the non-trivial $2 \times 2$ matrix is indeed the adjoint representation under SU(2). More explicitly, the covariant derivative of $\Phi$ is

$$D_\mu\Phi = \partial_\mu\Phi - ig(A\Phi + \Phi A^T)\,. \tag{A.24}$$

Focusing on the $2 \times 2$ matrix, this can be rewritten as

$$D_\mu\Phi \to D_\mu\Phi_{\text{SU(2)}} = \partial_\mu\Phi_{\text{SU(2)}} - ig(A_{\text{SU(2)}}\Phi_{\text{SU(2)}} - \Phi_{\text{SU(2)}}A_{\text{SU(2)}})\,, \tag{A.25}$$

---

[32]The singlets have constant field values because the covariant derivative just becomes the ordinary derivative for the singlet, and thus such a component must be a constant.

where $\Phi_{SU(2)}$ is the redefined Higgs field that behaves as the adjoint representation of SU(2), and $A_{SU(2)}$ is also the corresponding redefined gauge field,

$$
\begin{aligned}
\Phi_{SU(2)} &= \frac{2}{g\,r}\psi_1(r)\,x_i\,T^i_{SU(2)}, \\
T^i_{SU(2)} &= \frac{1}{2}\lambda^i \quad (i \in \{1,2,3\}), \\
A_{\mu\,SU(2)} &= \frac{1}{gr}\epsilon_{\mu ib}\,T^i_{SU(2)}\hat{r}_b\,.
\end{aligned}
\tag{A.26}
$$

Focusing on the non-zero (and $r$-dependent) monopole configuration, the action is now expressed in terms of $\Phi_{SU(2)}$ and $A_{\mu\,SU(2)}$ in the Bogomol'nyi limit as follows:

$$
\int d^4x \left[ -\frac{1}{4}F^a_{\mu\nu\,SU(2)}F^{a\mu\nu}_{SU(2)} + \frac{1}{2}(D_\mu\Phi^a_{SU(2)})(D^\mu\Phi^a_{SU(2)}) \right].
\tag{A.27}
$$

Here, $F^a_{\mu\nu\,SU(2)}$ denotes the field strength tensor for the SU(2) gauge field. The Bogomol'nyi equation is obtained just as in the case of SU(2) $\to$ U(1). The Bogomol'nyi equation is

$$
\frac{1}{g}B^a_{i\,SU(2)} - D_i\Phi^a_{SU(2)} = 0,
\tag{A.28}
$$

where $B^a_{i\,SU(2)} = -\frac{1}{2}\epsilon_{ijk}F^a_{ij\,SU(2)}$.

The dyon degree of freedom is related to the global U(1) transformation. From the above results, we can perform the following U(1) rotation,

$$
U' \equiv \exp\left(i\frac{\sigma}{v}\Phi_{SU(2)}\right) = \exp\left(i\frac{\sigma}{v}\frac{2\psi_1}{gr}\left(x\,T^1_{SU(2)} + y\,T^2_{SU(2)} + z\,T^3_{SU(2)}\right)\right).
\tag{A.29}
$$

Under this rotation, we have

$$
A_{SU(2)} \to U'A_{SU(2)}U'^{-1} + iU'(\partial U'^{-1}),
\tag{A.30}
$$

$$
\Phi_{SU(2)} \to U'\Phi_{SU(2)}U'^{-1} = \Phi_{SU(2)}.
\tag{A.31}
$$

Focusing on the non-trivial background configuration as in the previous case, this leads to

$$
\delta A^a_{i\,SU(2)} = \sigma\frac{1}{v}(\partial_i\Phi_{SU(2)})^a, \qquad \delta\Phi_{SU(2)} = 0.
\tag{A.32}
$$

Now, deriving the dyon action parallels the case of SU(2) $\to$ U(1). Focusing on the monopole background configuration, the $\theta$-term in the SU(3) gauge theory leads to

$$
\int d^4x\,\frac{1}{64\pi^2}G^a_{\mu\nu}G^a_{\rho\sigma}\epsilon^{\mu\nu\rho\sigma} \supset \int dt \int d^3x\,\frac{\theta}{8\pi^2}\dot{A}^a_{i\,SU(2)}B^a_{i\,SU(2)} \supset \frac{\theta}{2\pi}\dot{\sigma}.
\tag{A.33}
$$

Here, we used $D_iB^a_{i\,SU(2)} = 0$ and $\int d^2S_i\frac{1}{v}B^a_{i\,SU(2)}\Phi^a_{SU(2)} = 4\pi$. The $\theta$-term also includes another term by treating $A^a_{0\,SU(2)}$ as a constant,

$$
\int d^3x(-4D_iA^a_{0\,SU(2)})B^a_{i\,SU(2)} \supset -\theta\frac{1}{4\pi^2}\mathrm{Tr}[A_{0\,SU(2)}Q_M],
\tag{A.34}
$$

where we have $B^a_{i\,SU(2)} = Q_M\frac{\hat{r}_i}{4\pi r^2}$ for large $r$. Consequently, the dyon action from the $\theta$-term is expressed as

$$
\int \theta\frac{1}{2\pi}\left(d\sigma - \frac{2}{4\pi}\mathrm{Tr}[A_{SU(2)}Q_M]\right).
\tag{A.35}
$$

# B   Further examples

## B.1   $SU(5) \times SU(5) \rightarrow [SU(3) \times SU(2) \times U(1)]/\mathbb{Z}_6$

Next, let us consider an example in which $G$ is not simple, but semisimple, and simply connected. Specifically, consider the product gauge group $G = SU(5) \times SU(5)$. The product $SU(5) \times SU(5)$ is broken into $[SU(3) \times SU(2) \times U(1)]/\mathbb{Z}_6$ with the condensation of bi-fundamental fields [84]. The stable monopoles are generated due to the non-trivial homotopy group $\pi_2(G/H) \cong \pi_1(H) \cong \mathbb{Z}$.

In the UV regime, we identify two conserved UV CW currents associated with the product groups $SU(5) \times SU(5)$. On the other hand, the conserved IR current arises from only a linear combination of the UV currents. This conserved current results from the embedding of $[SU(3) \times SU(2) \times U(1)]/\mathbb{Z}_6$ within the diagonal $SU(5)$ subgroup of $SU(5) \times SU(5)$. It is important to note that further analysis is necessary when the gauge group involves a non-trivial quotient. For example, with the gauge group of $G = [SU(5) \times SU(5)]/\mathbb{Z}_5$, the fundamental group $\pi_1(G)$ is non-trivial, which leads to fractional monopoles. This situation requires a more detailed examination.

## B.2   $SU(5) \times U(1) \rightarrow SU(3) \times SU(2) \times U(1)$

Consider a final example in which the initial gauge group $G$ is neither simply connected nor semi-simple. The original group is $G = SU(5) \times U(1)$, which is broken into $H = SU(3) \times SU(2) \times U(1)$ by the Higgs in the **10** representation of $SU(5)$ with a $U(1)$ charge [85–88]. Note that $G$ and $H$ are direct products without any quotient, permitting fields to be solely charged under the $U(1)$ with arbitrary charges. The homotopy groups in this theory are $\pi_1(G) = \mathbb{Z}$ and $\pi_2(G/H) = 0$, implying no 't Hooft-Polyakov monopoles.

We find two UV CW currents associated with $SU(5)$ and $U(1)$. The CW current from $U(1)$ is recovered by the dyon degree of freedom. On the other hand, we identify three conserved IR CW currents because of the trivial homotopy group of $\pi_2(G/H) = 0$. Consequently, we encounter another example that the conserved IR CW currents do not correspond to the ones descending from the UV currents. As in the previous case, further study is needed when the quotient is non-trivial, e.g., $[SU(5) \times U(1)]/\mathbb{Z}_5 \rightarrow [SU(3) \times SU(2) \times U(1)]/\mathbb{Z}_6$.

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
