# Peer review of "Monopole Breaking of Chern-Weil Symmetries"

_SciPost Physics, doi:SciPost Phys. 18, 162 (2025)_

## Round 1 · Referee Report · Anonymous (Referee 1) · 2024-11-26

Report

In this manuscript, the authors explore the connection between the explicit breaking of $(d-3)$-form magnetic symmetries and the breaking of $(d-5)$-form instantonic symmetries. This builds on an intriguing strain of the authors' earlier work, including from the perspective of the effects of loops of magnetic monopoles, which had been crying out for further understanding. Various works utilizing generalized symmetries in particle physics have implicitly relied on this connection, but concrete understanding thereof had been absent from the literature. It is the $d=4$ case which is of course most useful for real-world physics, but this is also the most subtle case where the instantonic symmetry must be interpreted as a subtle sort of `$(-1)$-form symmetry', so this general study putting $d=4$ on roughly even footing with the higher-dimensional case is particularly valuable.

This is a great paper doing important field theory which needs to be understood in order to clearly see the impact of generalized symmetries on phenomenology. The investigation is thorough and contains useful explicit detail and calculation. The authors have put clear effort into providing pedagogical explanations of their investigation and results. The 2d toy models introduced in Section 2 are especially valuable as a simple setting in which the reader can get a handle on what is happening before having to discuss gauge theories. More-complicated scenarios are built up slowly, and the interesting features of each example are picked apart with lessons to be learned highlighted.

This work at the interface of formal and phenomenological field theory is a big, concrete step toward a full understanding of the breaking of Chern-Weil symmetries which will allow clearer use of these ideas in particle physics.

Recommendation

Publish (surpasses expectations and criteria for this Journal; among top 10%)

---

## Round 1 · Referee Report · Anonymous (Referee 2) · 2025-1-2

Report

This work discusses the breaking of (d-3)- and (d-5)-form symmetries by the dynamical monopoles and "restoration" of them by dyon zero modes living on the monopole world-volume. The authors provide both general theoretical descriptions as well as concrete examples and computations. The manuscript is well-written and the results are interesting and may have important impacts on both theoretical QFT and particle phenomenology. The quality of the work certainly meets the qualification for the publication and I think it deserves a publication.

I have a few questions and comments and I would like to invite the authors to clarify and/or update the paper if necessary.

  1. Suppose we have a QFT, with $G \to H$ breaking by a Higgs, which admits a set of ('t Hooft-Polyakov) monopole as well as dyons. The latter dyon solutions may be obtained by exciting dyonic zero modes associated with $U(1)$ symmetry as discussed in Appendix A of the paper. My question is if a theory contains both monopoles and dyons, then while dyons may respect (d-5)-form symmetry, monopoles will still break it. Therefore, in what sense can we say that dyonic zero modes allows a construction of an improved current, hence unbroken (d-5)-form symmetry? In other words, how does the dyonic zero modes, which in a sense make dyons not screen, eliminate the screening effects by monopoles (which do not have dyon zero mode excitations)?

  2. I found the discussion in section 4.4 useful but also slightly unclear (at least to me) in some places.

(2-1) While it said $w_2 \in H^2 (M, \mathbb{Z})$, based on what I understand, I feel that it instead should be a 2-form background gauge field for the $\mathbb{Z}_n^{(1)}$ electric center symmetry. Also, is eq(30) essentially a dualization of $w_2$ to its magnetic dual field?

(2-2) For (33), if $w_2$ is indeed a 2-form background gauge field for the $\mathbb{Z}_n^{(1)}$ electric center symmetry, then it will also appear in the gauge kinetic term $\sim \int {\rm tr} \left(g_2 - \frac{2\pi}{n} w_2 \right) \wedge * \left ( g_2 - \frac{2\pi}{n} w_2 \right)$. If this is true, then how do we still see that (33) is correct, which does not include terms from the gauge kinetic term in the evaluation of EoM for $w_2$? Also, I noticed that there is a sign flip between (33) and (34), i.e. what was $+$ between two terms in (33) becomes somehow $-$ in (34).

(2-3) In the sentence between (34) and (35), I believe by "genuine" you mean genuine line operator, correct? That is, in the absence of $\theta$-term, hence no Witten effect, we have a genuine 't Hooft line, but with $\theta \neq 0$, 't Hooft lines are dressed with Wilson surface. Is this what was meant? If so, I wonder if it'd be helpful to make this point slightly more explicit?

  1. Julia-Zee dyon vs Witten effect. Eq (15) is based on dyonic zero modes while (37) is from Witten effects. Do these two lead to the same effects when it comes to "restoring" (d-5) symmetry through the improved Chern-Weil current?

Recommendation

Publish (surpasses expectations and criteria for this Journal; among top 10%)

  • validity: high
  • significance: high
  • originality: top
  • clarity: top
  • formatting: excellent
  • grammar: perfect

Author:  Eduardo Garcia Valdecasas  on 2025-04-30  [id 5430]

(in reply to Report 2 on 2025-01-02)

We thank the referee for the positive review. The referee raised several points that deserved clarification, which we now provide.

  1. In theories that support dyons from the start (that is, dyons of integer electric charge) the story is unchanged. The crucial point is that, in the presence of a $\theta$-term, any object with magnetic charge will have dyonic modes that gives it a fractional electric charge. There are no ``bare’’ monopoles in such theories. The form of these dyonic d.o.f.~is the same for monopoles and dyons and can always be used to recover a conserved (d-5)-form symmetry.
  2. We thank the referee for raising this point. We agree that the discussion is unclear and we have modified it accordingly, hopefully addressing the concerns. In the following we answer the questions but we have reformulated the section, so we recommend the referee to see it in the new submission. 2.1 The referee correctly points out a typo. We modified to $w_2 \in H^2(M, \mathbb{Z}_n)$. Regarding the question about dualization: in the current manuscript, $\tilde B$ in Eq.(30) is introduced as a Lagrange multiplier, and this can be viewed as a dual description. 2.2 We agree with the referee that $w_2$ enters into the kinetic term, but we don't think it changes our results. In particular, the crucial coupling between $w_2$ and $\theta$ is unchanged. We have added footnote 14 and several comments on that section addressing this point. We corrected the minus signs. 2.3 With "genuine" we refer to a line operator that does not depend on the attached surface that may be used to define it. That is, correlation functions depend only on the boundary of the surface. Note that both in (33) and in (37) the operators are genuine by virtue of equations (32) and (36).
  3. Both Eq. (15) and Eq. (43) reflect the fact that the Witten effect implies that monopoles are accompanied by dyonic modes, and these modes play a crucial role in restoring the conservation of the improved Chern-Weil current. However, it is important to note that dyonic modes do not generically improve any Chern-Weil current. Their role becomes essential once the $\theta$-angle is promoted to a dynamical axion field. We have clarified this point in the vicinity of Sec. 4.4.1, in parallel with the discussion in the U(1) case.

---

## Round 2 · Author Response

We have replied to the referees in the previous submission.

---

## Round 2 · List of Changes

The list is provided in the replies to the referees.

---

## Editorial Decision

published